# Harnessing Heterogeneity: Improving Convergence Through Partial Variance Control in Federated Learning

## Abstract

Federated Learning (FL) has emerged as a promising paradigm for collaborative model training without sharing local data. However, a significant challenge in FL arises from the heterogeneous data distributions across participating clients. This heterogeneity leads to highly variable gradient norms in the model's final layers, resulting in poor generalization, slower convergence, and reduced robustness of the global model. To address these issues, we propose a novel technique that incorporates a gradient penalty term into partial variance control. Our method enables diverse representation learning from heterogeneous client data in the initial layers while modifying standard SGD in the final layers. This approach reduces variance in the classification layers, aligns gradients, and mitigates the effects of data heterogeneity. Through theoretical analysis, we establish convergence rate bounds for the proposed algorithm, demonstrating its potential for competitive convergence compared to current FL methods in highly heterogeneous data settings. Empirical evaluations on five benchmark datasets validate our approach, showing enhanced performance and faster convergence over state-of-the-art baselines across various levels of data heterogeneity. Our code is available at `https://anonymous.4open.science/r/FedPGVC-7F18`.

## 1 Introduction

Federated learning (FL) facilitates collaborative training of a global model across multiple clients while preserving data privacy by avoiding the need to transmit local data to a central server, in contrast to traditional centralized methods (McMahan et al., 2017). With the proliferation of decentralized data sources like mobile devices, hospitals, and the Internet of Things (IoT), FL has gained traction as a solution for training deep networks across distributed environments (Zhang et al., 2022). However, a significant practical obstacle encountered during federated training is data heterogeneity across clients (Kairouz et al., 2021; Li et al., 2020). Diverse user behaviors can lead to significant heterogeneity in the local data across different clients, resulting in non-independent and identically distributed (non-IID) data. This heterogeneity has been shown to cause unstable convergence, slow training progress, and ultimately suboptimal or even detrimental model performance (Li et al., 2022; Zhao et al., 2018). While FedAvg (McMahan et al., 2017) has been widely adopted and successful across multiple applications, it frequently encounters challenges in attaining optimal accuracy and convergence, particularly in heterogeneous data distributions. This difficulty arises from client drifts (Karimireddy et al., 2020), a phenomenon resulting from the varying nature of data among participating clients. Prior research has addressed the issue of client drift by introducing penalties for the divergence between client and server model (Li et al., 2020; 2021a), or by employing variance reduction approaches during the client model update (Karimireddy et al., 2020; Acar et al., 2021). Luo et al. (2021) tackled data heterogeneity through classifier re-training utilizing virtual features. Another study uncovers that a biased classifier significantly undermines the performance of federated training on heterogeneous data and introduces a novel algorithm by re-training the classifier with learnable features (Shang et al., 2022). A recent study by Li et al. (2023) measures gradient variability across clients by calculating drift diversity, especially in deeper layers, and proposes aligning classification layers using control variates. While this approach may enhance model performance, it can increase communication costs and relies on assumptions that may not always hold in practical FL scenarios.

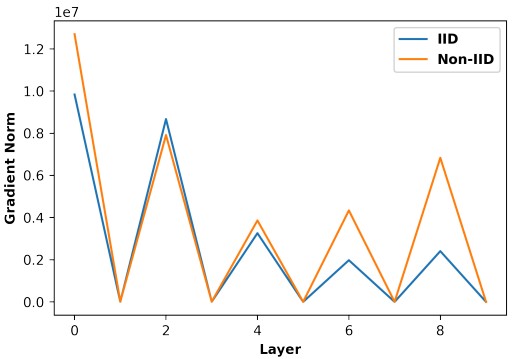

Figure 1: Comparison of gradient norm of the two models (IID and non-IID) trained using FedAvg.

Table 1: Aggregate metrics for gradient norms averaged across all layers of models trained on IID and non-IID data distribution.

| Metric | IID | non-IID |
|---|---|---|
| Mean Gradient Norm | $3.96 \times 10^6$ | $4.49 \times 10^6$ |
| Variance of Gradient Norm | $1.36 \times 10^13$ | $1.95 \times 10^13$ |
| Maximum Gradient Norm | $0.98 \times 10^7$ | $1.3 \times 10^7$ |

## 1.1 Empirical Observations

Based on these observations, we conducted two experiments: first, to empirically analyze gradient norms for models trained on IID and non-IID data, aiming to understand gradient behavior across layers and its impact on training stability; second, to determine which parts of the neural network are more sensitive to data heterogeneity. We utilized the CIFAR100 dataset and applied the Dirichlet distribution to simulate different data distributions. Specifically, we set the concentration parameter ($\alpha$) to 0.5 to create a non-IID distribution and to 100 for an IID distribution, employing a 5-layer CNN for both scenarios (Lin et al., 2020). Notably, smaller ($\alpha$) result in more skewed data distributions, effectively mimicking real-world scenarios where data is unevenly partitioned. The detailed experimental setting can be found in Subsection 5.1.

Initially, we calculate aggregate metrics, including the mean, variance, and maximum gradient norms across all layers of the CNN model trained on both IID and non-IID data distributions, averaging across all layers at the point of observed convergence. The results, presented in Table 1, reveal that models trained on non-IID data exhibit higher average gradient norms and greater variance compared to those trained on IID data. This indicates that non-IID training leads to larger updates and potentially greater instability in the training process. In the second experiment, we analyzed the gradient norms for each layer of the CNN model to understand the impact of distribution shifts. The results, shown in Fig. 1, illustrate the variation of gradient norms across layers for both IID and non-IID cases. Initial layers exhibit higher gradient norms, which decrease significantly in subsequent layers. Both models display similar gradient patterns in the initial layers. However, the model trained on non-IID data exhibits higher gradient norms in and near the classification layer, indicating larger updates and greater instability in these regions due to data heterogeneity. These findings highlight that the classification layer, along with its neighboring layers, significantly contributes to the observed instability and slower convergence when training with non-IID data. Note that the odd layers in Fig. 1 are the MaxPooling layers following convolutional layers, which lack trainable parameters and only downsample spatial dimensions by selecting the maximum value within each window. As these pooling layers do not participate in learning, their gradient norms are inherently zero.

Inspired by the above empirical observations, we propose Federated Partial Gradient Variance Control (FedPGVC) to stabilize noisy gradient norms to mitigate data heterogeneity without incurring additional communication costs. FedPGVC calculates a gradient penalty term for each individual client, updating the last layers of the neural network while the remaining layers are updated using a Stochastic Gradient Descent (SGD) optimizer. In this work, we calculate the gradient penalty term inspired by the Wasserstein Distributionally Robust Optimization (WDRO) (Gao & Kleywegt, 2023). This approach addresses distributional uncertainties and deviations, enhancing the global model's resilience to non-IID data across clients and improving generalization to unseen data samples, even when they deviate from the training distribution. We have performed experiments on the five widely used datasets, MNIST, FMNIST, CIFAR100, Tiny-ImageNet and QQP datasets, with varying degrees of data heterogeneity among clients. Our experimental results

demonstrate that the proposed FedPGVC requires fewer communication rounds to achieve the same level of accuracy as existing approaches. Furthermore, with a fixed number of communication rounds, FedPGVC attains comparable or superior top-1 accuracy. The key contributions of this work are as follows:

- We introduce FedPGVC to tackle the challenges of data heterogeneity in federated training by incorporating a partial variance reduction technique utilizing client-specific gradient penalty terms.

- We have proposed a gradient penalty term for the weight updates of the final classification layers to mitigate client drift, stabilize gradient diversity, and accelerate convergence in federated training.

- We offer a theoretical convergence for FedPGVC in both convex and non-convex scenarios, outlining its limited reliance on measures of data heterogeneity.

- Experimental analysis shows that the proposed FedPGVC surpasses prior state-of-the-art methods in both performance and convergence efficiency across different levels of data heterogeneity and a range of diverse datasets.

## 2 Related work

Numerous studies have explored effective strategies for addressing the challenges of data heterogeneity in FL. We have broadly categorized these approaches into three groups: 1) client drift mitigation, which adjusts the local objectives of clients to align their models more closely with the global model; 2) aggregation schemes, which enhance the server-side fusion mechanism for model updates; 3) personalized federated learning, which focuses on training personalized models for clients rather than a shared global model. In the proposed work, we mainly focused on techniques based on client drift mitigation.

FedAvg is a predominant optimization method in FL and has witnessed widespread adoption (McMahan et al., 2017). However, in heterogeneous settings where local objectives diverge significantly, FedAvg encounters performance degradation due to client drift, limiting its effectiveness in non-IID data scenarios (Karimireddy et al., 2020). Li et al. (2020) introduced a proximal regularization term to manage the divergence between client and server models but fails to align global and local optimal points effectively. Li et al. (2021b) employs local batch normalization (LBN) to mitigate feature shift before server-side model averaging. Sahoo et al. (2024a) introduces a novel loss function and an innovative way of calculating adaptive proximal term to tackle heterogeneous data settings. Additionally, it uses Self-organizing map (SOM) based server-side aggregation. Several techniques like FedBabu (Oh et al., 2021) and TCT (Yu et al., 2022) aim to enhance FL models by fine-tuning classifiers using standalone datasets or simulated features derived from client models. Similarly, Luo et al. (2021) addresses data heterogeneity by re-training classifiers with virtual features obtained from an approximated GMM model. MOON introduces a model-contrastive FL framework that aligns local client representations with the global model using a contrastive loss (Li et al., 2021a). It employs a momentum encoder to provide a stable target for the contrastive loss, acting as a temporal ensemble of the global model and mitigating client drift. Stochastic variance reduction based methods like SVRG (Johnson & Zhang, 2013), SAGA (Defazio et al., 2014), and their variations utilize control variates to mitigate the variance inherent in traditional SGD, enabling linear convergence rates for strongly convex optimization problems. SCAFFOLD (Karimireddy et al., 2020) and DANE (Shamir et al., 2014) have incorporated variance reduction techniques for the whole model on convex problems without exploring their performance in non-convex setups. Despite potential benefits, these approaches incur higher communication costs due to transmitting additional control variates, posing challenges for resource-constrained IoT devices (Halgamuge et al., 2009). Additionally, the existing methods have shown rapid convergence in simpler models, and their effectiveness on deep networks, remains largely unexplored (Sahoo et al., 2024b). FedPVR (Li et al., 2023) offers a novel perspective on FedAvg's performance in deep neural networks, uncovering substantial heterogeneity in client-specific final classification layers. By introducing targeted variance reduction exclusively for these last layers, FedPVR achieves remarkable improvements over established benchmarks. Motivated by the previous observations, our research focuses on improving the partial variance control of individual clients to mitigate data heterogeneity problems.

## 3 Method

### 3.1 Problem Statement

The primary aim of this work is to develop a robust model that can learn collaboratively from decentralized clients without the need for data sharing. The focus is on enhancing performance during federated training, particularly in scenarios with non-IID data. Given $K$ clients, where each client $k \in \{1, \ldots, K\}$ possesses a local dataset $D_k$, the aim is to learn a generalized global model over $D = \bigcup_{k=1}^{K} D_k$. The global objective function is defined as:

$$\arg \min_{w} L(w) = \sum_{k=1}^{K} \frac{|D_k|}{|D|} L_k(w), \tag{1}$$

where the local objective function $L_k(w)$ for client $k$ measures the local empirical loss over the data distribution $D_k$ and is given by:

$$L_k(w) = \mathbb{E}_{x \sim D_k}[\ell_k(w; x)], \tag{2}$$

here, $\ell_k$ is the loss function for client $k$, while $w$ denotes the global model parameters to be optimized. This work emphasizes addressing the issue of data heterogeneity in FL due to the non-IID distribution of data across clients.

### 3.2 Theoretical Analysis

As mentioned in the introduction, non-IID data distributions among clients in federated learning environment result in increased gradient diversity, particularly in the last layers of the network. To tackle this challenge, we suggest a straightforward but powerful enhancement to the standard FedAvg algorithm. Our approach introduces a carefully designed gradient penalty term with the standard SGD to align gradient norms effectively in the final layers. This method not only reduces the effects of client data heterogeneity but also enhances model performance and accelerates convergence. In the FL framework, each client $k$ is associated with a local dataset $D_k$ and computes the gradient of the loss function with respect to the model parameters $w'$. The local loss function for client $k$ is defined by Eq. 3:

$$L_k(w') = \frac{1}{|D_k|} \sum_{i \in D_k} \ell(w'; x_i, y_i), \tag{3}$$

where $\ell(w'; x_i, y_i)$ is the loss for sample $(x_i, y_i)$. In the FedAvg algorithm, the global model parameters are updated according to the Eq. 4:

$$w_{t+1} = w_t - \eta_g \frac{1}{K} \sum_{k=1}^{K} \Delta w'_k, \tag{4}$$

where $\eta_g$ is the global learning rate, $\Delta w_k = \nabla L_k(w_t)$ is the local update from client $k$ and $w_t$ is the global model parameter for round $t$.

In IID settings, the data distribution remains consistent across all clients, leading to similar gradient norms. Let $\sigma_{iid}$ represent the standard deviation of gradient norms in IID settings presented in Eq. 5:

$$E[||\nabla L_{\text{iid}}(w)||] = \sigma_{\text{iid}}. \tag{5}$$

In non-IID settings, where data distributions vary across clients, we assume that gradient norms exhibit higher variability compared to IID settings, as presented in Eq. 6:

$$E[||\nabla L_{\text{non-iid}}(w)||] = \sigma_{\text{non-iid}} \gg \sigma_{\text{iid}}. \tag{6}$$

Similarly, we assume that the deeper layers of a neural network are responsible for learning more specific features, resulting in higher gradient norms in non-IID settings, as illustrated in Fig. 1 and presented in Eq. 7:

$$E[||\nabla L_{l,\text{non-iid}}||] \gg E[||\nabla L_{l,\text{iid}}||], \tag{7}$$

where $l$ denotes the index for the last layers. To address this gradient diversity, we propose to use a client-specific term called gradient penalty ($\rho_i$) for client $i$ to ensure better alignment of the gradients of the last layers of the model as presented in Eq. 8:

$$\Delta w_{t+1,l} = w_t - \eta_l \rho_l \nabla L_l, \tag{8}$$

where $\rho_l$ reduces gradient norm variations, and $\eta_l$ is the local learning rate. We choose $\rho_l$ as presented in Eq. 9:

$$\rho_l = \frac{\sigma_{\text{iid}}}{||\nabla L_{l,\text{non-iid}}||}. \tag{9}$$

This ensures that the gradients are aligned to resemble those in IID settings, as shown in Eq. 10:

$$||\rho_l \nabla L_{l,\text{non-iid}}|| = \sigma_{\text{iid}}. \tag{10}$$

### 3.3 Proposed Method

Motivated by these empirical and theoretical observations, we introduce FedPGVC, an innovative method for managing data heterogeneity in federated learning. Our algorithm (Algo. 1) includes three main components: i) client update (Eq. 13 and Eq. 15), ii) computation of client gradient penalty term (Eq. 14), and iii) server update (Eq. 16). Let $e \in \{0,1\}^Z$ be a binary vector, where each element $e_j$ indicates whether the corresponding layer is included in the partial gradient variance control. The sum $\sum e$ represents the number of selected layers to apply gradient variance control (refer to Eq. 11). This vector serves as a mask to differentiate between the initial layers of the model and those adjacent to or comprising the classifier. For the subset of indices $j$ where $e_j = 1$ (denoted as $S_{\text{gvc}}$ in Eq. 12), we modify the corresponding weights $y_{(i,S_{\text{gvc}})}$ to minimize variance. This is achieved by introducing a client-specific gradient penalty term $\rho_i \in \mathbb{R}^v$ as formulated in Eq. 14. Subsequently, we update the weights of the corresponding layer using Eq. 15. For the remaining indices, denoted as $S_{\text{sgd}}$, we update the corresponding weights $y_{(i,S_{\text{sgd}})}$ using standard SGD as formulated in Eq. 13. In each communication round, the process unfolds as follows: Every client receives a copy of the server model, denoted as $w$. Subsequently, each client independently executes $E$ model updating steps, leveraging the cross-entropy loss function as the optimization objective. These updating steps are governed by the equations (refer to Eq. 13, Eq. 14, and Eq. 15), which encapsulate the core operations involved in a single step. Once the local model updates are completed, the clients transmit their updated models, represented as $y_i$, back to the server. The server then aggregates these individual client models through the aggregation mechanism defined in Eq. 16.

$$e \in \{0,1\}^Z, \quad v = \sum_{j=1}^{Z} e_j \tag{11}$$

$$S_{\text{gvc}} := \{j : e_j = 1\}, \quad S_{\text{sgd}} := \{j : e_j = 0\} \tag{12}$$

$$y_{(i,S_{\text{sgd}})} \leftarrow y_{(i,S_{\text{sgd}})} - \eta_l g_i\left(y_{(i,S_{\text{sgd}})}\right) \tag{13}$$

$$\rho_i \leftarrow \frac{1}{B} \sum_{b=1}^{B} \ell(\theta, x_b) \nabla_\theta \ell(\theta, x_b) \tag{14}$$

$$y_{(i,S_{\text{gvc}})} \leftarrow y_{(i,S_{\text{gvc}})} - \eta_l * \rho_i * g_i\left(y_{(i,S_{\text{gvc}})}\right) \tag{15}$$

$$w \leftarrow (1 - \eta_g)w + \frac{1}{K}\sum_{i \in K} y_i \qquad (16)$$

Here, $B$ is the batch size, $\ell(\theta, x_b)$ is the loss function evaluated on the $b^{th}$ data point $x_b$ with model parameters $\theta$, $\nabla_\theta \ell(\theta, x_b)$ is the gradient of the loss function with respect to the model parameters $\theta$, evaluated on the $b^{th}$ data point $x_b$. $g_i(\theta)$ is defined as $g_i(\theta) := \nabla f_i(\theta; \zeta_i)$, where $g_i(\theta)$ is an unbiased gradient of local objective of $i^{th}$ client $f_i$ with its variance bounded by $\sigma^2$, represented as $\mathbb{E}[g_i(x)] = \nabla f_i(x)$ and $\mathrm{Var}(g_i(x)) \leq \sigma^2$. $\zeta_i$ represents a random variable, allowing $g_i(\theta)$ to serve as an unbiased estimate of the true gradient of the overall objective function.

Note that $\sigma_{iid}$ in Eq. 9 specifically addresses gradient diversity in the final layers by defining as a scaling factor that normalizes non-IID gradients to align with IID gradients, thus establishing an ideal definition. This normalization stabilizes gradient norms across clients in the layers most sensitive to heterogeneity, enhancing model alignment and mitigating client drift. Building on this, Eq. 14 extends the gradient penalty term for practical applications by defining it as an average of gradients over each client's batch. This formulation adapts to local data distributions, enabling flexible variance control in the last layers where the non-IID effect is pronounced, capturing client-specific data characteristics. In addition, clients exhibiting high statistical heterogeneity, that is, those whose local data distributions significantly diverge from the global model, tend to generate gradients that deviate markedly from the global update direction while also exhibiting higher variance than those of more homogeneous clients. Equation 14 addresses this challenge by incorporating a gradient penalty term that dynamically scales and increases the weight of these high-divergence gradients, ensuring that underrepresented distributions are not overwhelmed during the server's averaging process and ultimately allowing the model to generalize better across all clients.

---

**Algorithm 1** Federated Partial Gradient Variance Control (FedPGVC)

---

1: **Server:** Initialize the global model parameters $w^0$, Global learning rate $\eta_g$.
2: **Client:** Initialize the local model parameters $y_i^0$, Local learning rate $\eta_l$.
3: Define a mask $e \in \{0, 1\}^Z$, where $e_j = 1$ for the last few layers and 0 for the rest layers.
4: Let $S_{\mathrm{sgd}} = \{j : e_j = 0\}$ and $S_{\mathrm{gvc}} = \{j : e_j = 1\}$.
5: **for** $r = 1, 2, \ldots, R$ **do**
6:     Server broadcasts the global model $w^0$ to all clients.
7:     **for** each client $i = 1, 2, \ldots, K$ in parallel **do**
8:         **for** $\phi = 1, 2, \ldots, E$ **do**
9:             $y_{(i,S_{\mathrm{sgd}})}^{(r,\phi)} = y_{(i,S_{\mathrm{sgd}})}^{(r,\phi-1)} - \eta_l \nabla_{S_{\mathrm{sgd}}} f_i(y_i^{(r,\phi-1)})$
10:            $\rho_i^{r-1} \leftarrow \frac{1}{B}\sum_{b=1}^B \ell(\theta, x_b)\nabla_\theta \ell(\theta, x_b)$
11:            $y_{(i,S_{\mathrm{gvc}})}^{(r,\phi)} = y_{(i,S_{\mathrm{gvc}})}^{(r,\phi-1)} - \eta_l \nabla_{S_{\mathrm{gvc}}} L_i(y_i^{(r,\phi-1)}, \rho_i^{r-1})$
12:         **end for**
13:         Client $i$ sends the updated model $y_i^{(r,E)}$ to the server.
14:     **end for**
15:     Server aggregates the client models and updates the global model:
16:     $w^r = (1 - \eta_g)w^{(r-1)} + \frac{1}{K}\sum_i y_i^{(r,E)}$
17: **end for**

---

### 3.3.1 Usefulness of Introducing Gradient Penalty ($\rho$)

The intuition behind introducing the gradient penalty term $\rho$ into the standard SGD for the last layers of the model is to encompass both the direction and strength of the gradients, along with the loss landscape for each client's data distribution. Prioritizing the gradients from clients which have data distributions that significantly deviate from global distribution allows us to better handle the most challenging situations within a specific range around each client's observed data distribution. Incorporating $\rho$ into the weight updates for the final classification layers of the neural network allows us to achieve better alignment of these layers across clients, mitigating the issue of client drift caused by data heterogeneity. Specifically, we update the

weights of the classification layers as presented in Eq. 14. This approach has the advantage of not requiring any additional communication overhead like prior methods such as SCAFFOLD and FedPVR (Karimireddy et al., 2020; Li et al., 2023). Moreover, it strikes a balance between diversity and uniformity across the layers of the neural network, allowing the feature extraction layers to learn rich representations while ensuring better alignment of the final layers across clients.

## 4  Convergence Proof

In this section, we provide the convergence analysis of the proposed FedPGVC, considering both convex and non-convex scenarios. To enable a theoretical proof, we introduce the following notations and assumptions: we consider K clients, each linked to a local objective function $f_i(x)$, where $i = 1, \ldots, K$. We impose the following assumptions on the objective functions:

**Assumption 1: Lipschitz smoothness:**

(a) $|\nabla f_i(x) - \nabla f_i(y)| \leq L|x - y|$ for all $x, y$ and some constant $L > 0$.

(b) $F(x) \leq F(y) + \langle \nabla F(y), x - y \rangle + \frac{L}{2}\|x - y\|^2$, where $F(x)$ is the global objective function.

**Assumption 2: Bounded gradients:** $\mathbb{E}|\nabla f_i(x; \xi_i)|^2 \leq G^2$ for all $x$ and some constant $G > 0$, where $\xi_i$ denotes the random variable representing the data samples used to compute stochastic gradients on client $i$.

**Assumption 3: Non-convexity:** We have first considered the non-convex setting, which is more general and applicable to deep neural networks. Additionally, we assume a measure of data heterogeneity across clients $\hat{\zeta}^2$ such that:

$$\frac{1}{K}\sum_{i=1}^{K}\mathbb{E}|\nabla f_i(x; \xi_i)|^2 \leq \hat{\zeta}^2, \quad \forall x \tag{17}$$

**Assumption 4: Bounded Variance**: The variance of the stochastic gradients on each client is bounded by $\sigma^2$, i.e.,

$$\mathbb{E}\left[\|\nabla f_i(y) - \nabla F(x)\|^2\right] \leq \sigma^2, \quad \forall i, x, y.$$

The above assumption 1,2, and 4 are aligned with those presented in (Li et al., 2020), (Wang et al., 2020), (Durmus et al., 2021) and (Nguyen et al., 2018) respectively. The convexity and non-convexity assumptions align with those presented in (Karimireddy et al., 2020).

**Theorem 1:**

Let $F(x) = \frac{1}{K}\sum_{i=1}^{K} f_i(x)$, with $F^*$ being the optimum value of F. For any $L$-smooth function $f_i$, the output of FedPGVC with $\eta_g = \sqrt{K}$ and suitable values of $\eta_l$ and number of FL rounds ($R$) satisfies:

**Non-Convex:**

$$\frac{1}{R}\sum_{r=0}^{R-1}\mathbb{E}\left[\|\nabla F(x^{(r)})\|^2\right] = O\left(\frac{1}{\sqrt{ER}}\right) + O\left(\frac{\sigma^2}{\sqrt{ER}}\right). \tag{18}$$

**Convex:**

$$\frac{1}{R}\sum_{r=0}^{R-1}\mathbb{E}[F(x^{(r)})] - F^* = \mathcal{O}\left(\frac{G\sqrt{K}}{\sqrt{KR}} + \frac{\hat{\zeta}\sqrt{E}}{\sqrt{R}} + \frac{\hat{\zeta}_p\sqrt{E}}{\sqrt{KR}} + \frac{F(x^{(0)}) - F^*}{R}\right). \tag{19}$$

where $\hat{\zeta}_p$ is a measure of the heterogeneity of the gradients for the layers where modified SGD is applied, which is defined in Eq. 30. The proofs for Theorem 1 can be found in Section 6 of Appendix. By setting explicitly mask $e = 0$ enforces $\hat{\zeta}_p^2 = 0$, effectively removing the gradient penalty term for the last layers and applying standard SGD across all layers. This reduction yields the standard FedAvg algorithm, where each client's model update relies solely on local gradients without additional variance control, making FedAvg a specific instance of FedPGVC with no gradient stabilization.

# 5 Experimental results

## 5.1 Experimental setup

To evaluate the efficacy of FedPGVC, we conducted comprehensive experiments using five widely recognized classification benchmarks: Tiny-ImageNet (Le & Yang, 2015), MNIST (LeCun et al., 2010), FMNIST (Xiao et al., 2017), CIFAR100 (Krizhevsky, 2009), and Quora Question Pairs (QQP) from GLUE benchmark (Wang et al., 2018). To ensure robustness and reliability, all experiments were conducted thrice using distinct seeds. We report the average of the maximum test accuracy obtained with each seed and its standard deviation, following the methodology described in Xu et al. (2022).

We partitioned the entire dataset client-wise using a strategy inspired by Lin et al. (2020) to create a real-world non-IID distribution. This was achieved by distributing the data among clients using a Dirichlet distribution with a concentration parameter $\alpha$, which can take any real positive value. The measure of data heterogeneity across clients is governed by the $\alpha$ with a smaller value, resulting in a more skewed data distribution, mimicking real-world scenarios where data is unevenly partitioned. Figure 9 in the appendix illustrates an example of such a non-uniform data distribution for the MNIST dataset. In our experiments, we adopted $\alpha$ values of 0.5 and 1.0, which are commonly employed values (Lin et al., 2020) to simulate varying levels of data heterogeneity. Each client possesses its own local data partition, which remains unchanged throughout the communication rounds. This static data distribution allows us to access the performance of our proposed method under realistic conditions where clients do not exchange data. To assess the classification performance of the global model, we hold out a test dataset at the server, which remains unseen during the training process. For our experiments, we utilized the well-established LeNet (LeCun et al., 1998) neural network for the MNIST and FMNIST datasets. For CIFAR-100, we employed a 5-layer CNN following the approach described in (Duan et al., 2023), ResNet18, and viT (ViT-B/32). For the Tiny-ImageNet dataset, we employed the ResNet18 architecture, while for the QQP dataset, we implemented a straightforward two-layer LSTM network. We applied the variance reduction technique to the last two layers of the selected models to address data heterogeneity. Our experimental setup involved 10 participating clients in each communication round, with a batch size of 32, consistent with the configurations reported in prior studies (Li et al., 2023) and (Yu et al., 2022). In our experimental setup, each client performed two local epochs of model updating. Consistent with the configuration outlined in (Karimireddy et al., 2020), we fixed the server learning rate $\eta_g = 1$. To determine the optimal client learning rate for each experiment, we conducted a grid search over $0.05, 0.01, 0.2, 0.3$. Our implementation of FedProx involved testing a range of proximal values $0.001, 0.1, 0.4, 0.7$ to determine the optimal setting. For FedNova, we selected the best proximal SGD value from the set $0.001, 0.003, 0.05, 0.1$, in accordance with the recommendations in (Li et al., 2024). Across all experiments, we employed the Adam optimizer for consistency.

Table 2: Average of best Test Accuracies (%) with standard deviation on MNIST, FMNIST, and CIFAR100 datasets with varying degrees of data heterogeneity. The values in bold represent the highest accuracy achieved. Standard deviation values are provided in parentheses.

| | MNIST | | FMNIST | | CIFAR100 | |
|---|---|---|---|---|---|---|
| | $\alpha = 0.5$ | $\alpha = 1.0$ | $\alpha = 0.5$ | $\alpha = 1.0$ | $\alpha = 0.5$ | $\alpha = 1.0$ |
| Fedavg | 99.00 ($\pm$0.03) | 98.89 ($\pm$0.05) | 88.65 ($\pm$0.22) | 89.14 ($\pm$0.18) | 24.25 ($\pm$0.16) | 25.00 ($\pm$0.14) |
| FedProx | 98.99 ($\pm$0.04) | 99.02 ($\pm$0.03) | 86.96 ($\pm$0.30) | 89.10 ($\pm$0.21) | 24.89 ($\pm$0.15) | 25.56 ($\pm$0.13) |
| FedNova | 98.91 ($\pm$0.05) | 98.79 ($\pm$0.06) | 87.52 ($\pm$0.25) | 88.82 ($\pm$0.23) | 22.29 ($\pm$0.19) | 24.92 ($\pm$0.15) |
| FedBN | 98.94 ($\pm$0.04) | 99.03 ($\pm$0.04) | 88.76 ($\pm$0.20) | 89.17 ($\pm$0.18) | 25.12 ($\pm$0.12) | 25.66 ($\pm$0.13) |
| SCAFFOLD | 98.95 ($\pm$0.05) | 98.95 ($\pm$0.05) | 87.98 ($\pm$0.28) | 88.41 ($\pm$0.22) | 24.30 ($\pm$0.17) | 25.27 ($\pm$0.16) |
| FedPVR | 98.93 ($\pm$0.04) | 98.99 ($\pm$0.05) | 87.28 ($\pm$0.31) | 88.37 ($\pm$0.26) | 20.59 ($\pm$0.25) | 17.57 ($\pm$0.20) |
| **Proposed** | **99.05 ($\pm$0.02)** | **99.04 ($\pm$0.03)** | **88.83 ($\pm$0.18)** | **89.35 ($\pm$0.17)** | **25.29 ($\pm$0.11)** | **25.74 ($\pm$0.12)** |

## 5.2 Comparison with the State-of-the-art Methods

We evaluate our proposed FedPGVC against several notable FL algorithms, including FedAvg, FedProx, FedNova, FedBN, SCAFFOLD and FedPVR, and reported the results in Table 2. Across MNIST, FMNIST, and CIFAR-100, FedPGVC consistently achieves superior accuracy under varying heterogeneity levels ($\alpha =$

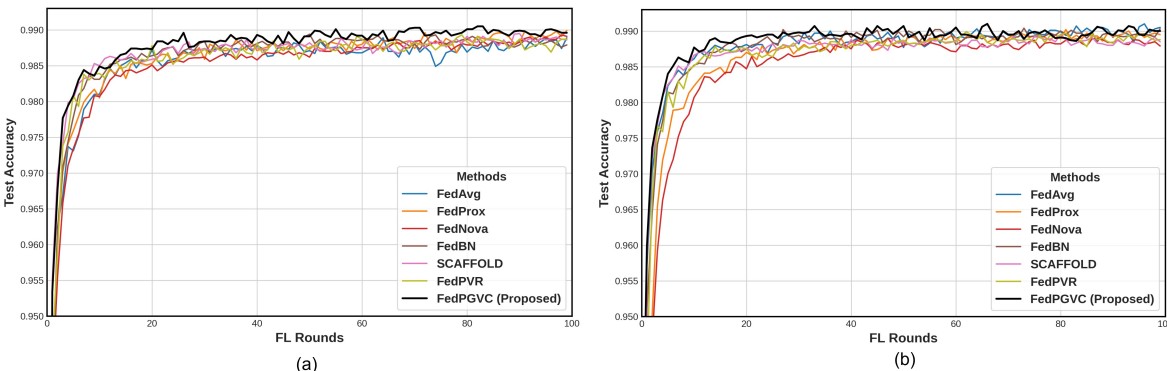

(a)   (b)

Figure 2: Performance comparison of proposed FedPGVC with baseline approaches: (a) and (b) depict the graphs on the MNIST dataset for $\alpha = 0.5$ and 1.0 respectively.

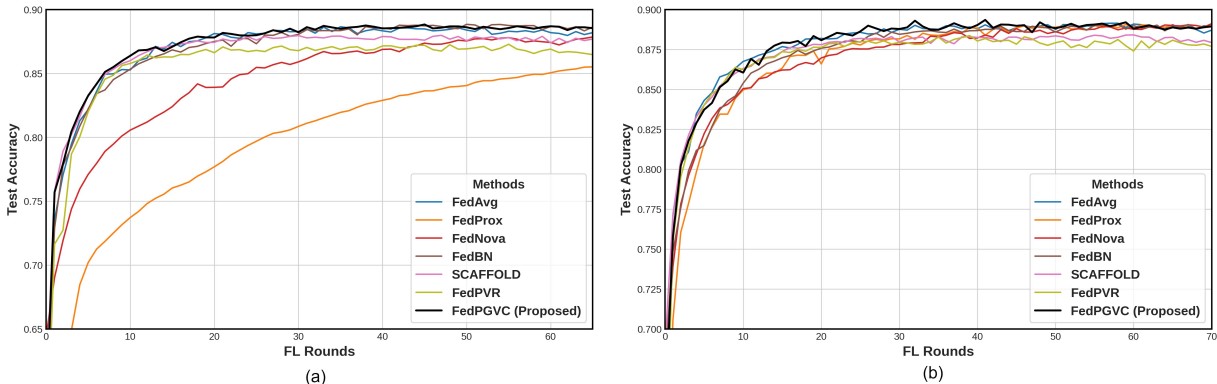

(a)   (b)

Figure 3: Performance comparison of proposed FedPGVC with baseline approaches: (a) and (b) depict the graphs on the FMNIST dataset for $\alpha = 0.5$ and 1.0 respectively.

0.5 and 1), achieving gains from as little as 0.01% to as high as 8.17% over baselines. While our method consistently outperforms baseline approaches across all datasets, the performance gain on MNIST is less pronounced. This can be attributed to the fact that MNIST is a relatively simple and well-studied dataset with low intraclass variability and minimal noise. As a result, most modern FL algorithms already achieve near-optimal accuracy on MNIST, leaving little room for further improvement. The stronger performance gains on more complex datasets such as CIFAR-100, which exhibit higher intraclass variability and are more susceptible to the challenges of data heterogeneity, further highlight the strengths of our approach. Please note that the lower classification accuracy on the CIFAR100 dataset is due to the use of a simple 5-layer CNN and the introduction of severe data heterogeneity in our experimental setup.

### 5.3   Convergence Analysis

Figures 2, 3, and 4 show that FedPGVC consistently outperforms baselines across MNIST, FMNIST, and CIFAR100. The corresponding graphs with error bars are provided in the Appendix (Fig.12, 13, and 14). FedPGVC achieves near 99% accuracy on MNIST within 23–27 rounds, about 88% on FMNIST in 15–18 rounds, and higher final accuracy on CIFAR100. As summarized in Table 3, FedPGVC converges 1.1–4.3× faster, owing to its effective variance reduction mechanism.

### 5.4   Experiments on Large Datasets and with various Backbones

To validate the effectiveness and generalizability of the proposed method, we employed more complex models, including ResNet18 (He et al., 2016) and Vision Transformer (ViT) (ViT-B/32) (Dosovitskiy, 2020), and

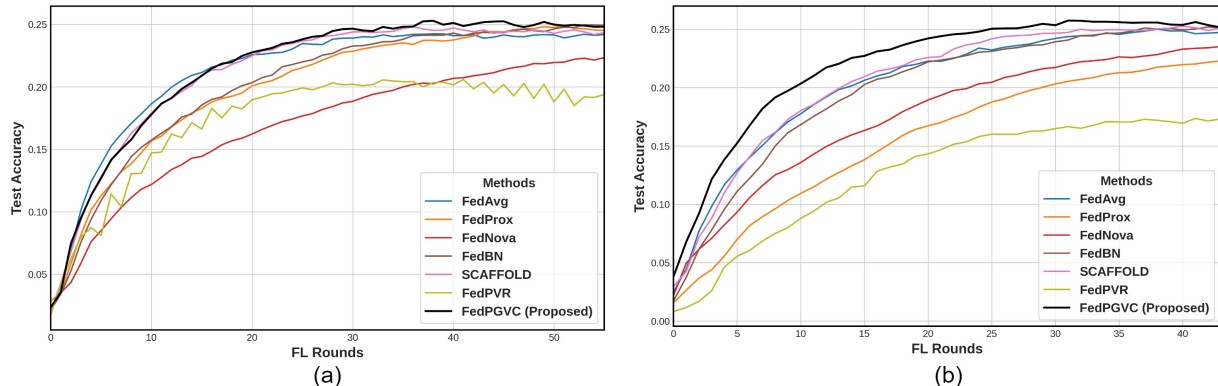

Figure 4: Performance comparison of proposed FedPGVC with baseline approaches: (a) and (b) depict the graphs on the CIFAR100 dataset for $\alpha = 0.5$ and 1.0 respectively.

Table 3: Number of communication rounds required (speedup compared to FedAvg) to achieve specific top-1 accuracy levels (99% for MNIST, 88% for FMNIST and 24% for CIFAR100). FedPGVC outperforms other methods by requiring fewer rounds to achieve comparable accuracy. '*' denotes the algorithm failed to achieve given test accuracy.

|  | MNIST | | FMNIST | | CIFAR100 | |
|---|---|---|---|---|---|---|
|  | $\alpha = 0.5$ | $\alpha = 1.0$ | $\alpha = 0.5$ | $\alpha = 1.0$ | $\alpha = 0.5$ | $\alpha = 1.0$ |
|  | Number of rounds | | Number of rounds | | Number of rounds | |
| FedAvg | 100 (1.0x) | 100 (1.0x) | 20 (1.0x) | 20 (1.0x) | 30 (1.0x) | 30 (1.0x) |
| FedProx | 77 (1.2x) | 74 (1.3x) | * | 28 (0.7x) | 43 (0.69x) | * |
| FedNova | 100 (1.0x) | * | * | 34 (0.5x) | * | * |
| FedBN | 50 (2.0x) | 28 (3.5x) | 26 (0.7x) | 24 (0.8x) | * | 33 (0.90x) |
| SCAFFOLD | * | * | * | 25 (0.8x) | 30 (1.0x) | 30 (1.0x) |
| FedPVR | * | * | * | 22 (0.9x) | * | * |
| **Proposed** | **23 (4.3x)** | **27 (3.7x)** | **18 (1.1x)** | **15 (1.3x)** | **27 (1.1x)** | **20 (1.5x)** |

utilized the Tiny-ImageNet dataset. Additionally, we included a language understanding task from the GLUE benchmark (Wang et al., 2018), specifically the QQP dataset, following (Sun et al., 2024). All experiments are conducted with $\alpha = 0.5$ and the results are reported in Table 8. For both ResNet18 and ViT, we added two linear layers and a classification layers on top of the pre-trained models. For the language understanding task, we used an LSTM network with two LSTM layers, two linear layers, and a final classification layer. For all the experiments, we maintain the same experimental settings as in the main experiments. On the CIFAR-100 dataset, FedPGVC achieves a minimum improvement of 0.48% over FedProx and a maximum of 5.74% over FedPVR using the ResNet18 model. In contrast, the ViT implementation shows a minimum improvement of 0.90% compared to FedAvg and SCAFFOLD, with a maximum improvement of 4.11% over FedPVR. On the Tiny-ImageNet dataset, the proposed method yields a minimum improvement of 0.92% over FedPVR and a maximum of 2.33% over FedNova. Additionally, for the language understanding task, our method outperforms the baselines with a minimum improvement of 0.72% over SCAFFOLD and a maximum of 2.90% over FedPVR.

## 5.5 Effect of Partial Gradient Variance Control (PGVC)

We integrated the proposed PGVC with standard FL approaches, and the results are summarized in Table 5. Additionally, extended results with standard deviations are provided in Table 6 in the Appendix section. The table shows that most approaches incorporating PGVC on the client side improved overall accuracy across datasets. FedAvg with PGVC achieves a 0.18% improvement on the FMNIST dataset and a 1.04%

Table 4: Performance of the proposed approach across various complex models on the CIFAR-100 and Tiny-ImageNet datasets, along with a language understanding task using the QQP dataset.

|  | CIFAR100 (ResNet18) | CIFAR100 (ViT) | Tiny-ImageNet (ResNet18) | QQP (LSTM) |
|---|---|---|---|---|
| Fedavg | $33.37 \pm 0.17$ | $23.44 \pm 0.11$ | $27.26 \pm 0.20$ | $62.80 \pm 0.20$ |
| FedProx | $33.71 \pm 0.09$ | $22.60 \pm 0.07$ | $27.61 \pm 0.23$ | $62.20 \pm 0.34$ |
| FedNova | $33.00 \pm 0.07$ | $21.28 \pm 0.03$ | $26.96 \pm 0.14$ | $61.32 \pm 0.52$ |
| FedBN | $33.48 \pm 0.12$ | $22.64 \pm 0.06$ | $28.31 \pm 0.25$ | $63.10 \pm 0.21$ |
| SCAFFOLD | $33.52 \pm 0.10$ | $23.44 \pm 0.02$ | $28.35 \pm 0.22$ | $62.99 \pm 0.36$ |
| FedPVR | $28.45 \pm 0.15$ | $20.23 \pm 0.03$ | $28.37 \pm 0.13$ | $60.81 \pm 0.44$ |
| **Proposed** | $\mathbf{34.19 \pm 0.08}$ | $\mathbf{24.34 \pm 0.04}$ | $\mathbf{29.29 \pm 0.20}$ | $\mathbf{63.71 \pm 0.41}$ |

improvement on CIFAR100. Similarly, FedPGVC enhances accuracy with FedProx and FedNova. However, FedBN with PGVC results in lower accuracy than FedBN alone, likely due to conflicting effects on training dynamics and model regularization. These findings demonstrate that the proposed PGVC approach can be effectively integrated with existing FL algorithms. For a comprehensive view of model convergence, we have presented the learning curves in Fig. 10 and Fig. 11 of the Appendix. In terms of computational efficiency, our proposed FedPGVC method requires 18 minutes for the CIFAR100 training, compared to 12 minutes for standard FedAvg. Similarly, on FMNIST, the training time increases from 15 to 21 minutes. While this represents a modest increase in computational cost, we argue that the significant accuracy gains justify this trade-off, especially in scenarios where model performance is paramount.

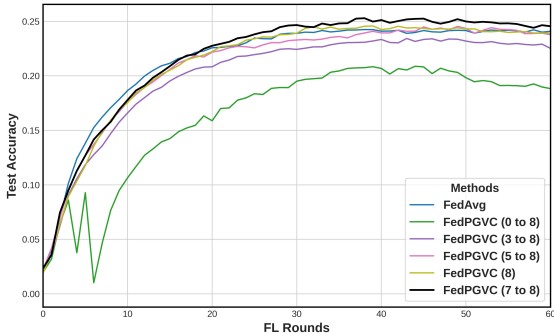

Figure 5: Performance of applying partial gradient variance control on different layers of the CNN model on the CIFAR100 dataset with $\alpha = 0.5$.

Table 5: The effect of applying the proposed method to existing popular baselines on the FM-NIST and CIFAR100 dataset with $\alpha = 0.5$.

| Method | FMNIST | CIFAR100 |
|---|---|---|
| FedAvg | 88.65 | 24.25 |
| FedAvg + PGVC | **88.83** (0.18 ↑) | **25.29** (1.04 ↑) |
| FedProx | 86.96 | **24.89** |
| FedProx + PGVC | **88.07** (1.11 ↑) | 24.58 (0.31 ↓) |
| FedBN | **88.86** | **25.12** |
| FedBN + PGVC | 88.11 (0.75 ↓) | 23.86 (1.26 ↓) |
| FedNova | 87.52 | 22.29 |
| FedNova + PGVC | **87.87** (0.35 ↑) | **24.82** (2.53 ↑) |

## 5.6 Applying PGVC on the Different Layers of the Model

Given that the proposed approach partially applies the gradient variance control technique in the last layers of the neural network, we investigated the effects of incorporating variance reduction in different layers. We conducted experiments on the CIFAR100 dataset with $\alpha = 0.5$, as shown in Fig. 5. The corresponding graph with error bars is provided in the Appendix (Fig. 15(a)) The results indicate that initiating variance reduction in the final layers of the model facilitates faster convergence and achieves the highest top-1 accuracy. Activating variance control in layers closer to the classifier yields minimal performance impact, whereas applying the technique in the initial layers leads to significant degradation. This observation is further validated by the use of partial variance control exclusively in the final layer, supporting our hypothesis that variance reduction is primarily required in the later layers of the network. Preserving diversity in the middle and early layers enables the learning of rich feature representations while promoting uniformity in the classifier layers, which helps make less biased decisions. This balance is crucial for leveraging the collective knowledge of distributed models while mitigating the adverse effects of excessive variance. We

have conducted the same experiment on the FMNIST dataset with $\alpha = 0.5$, and the results are presented in Section 7 of the Appendix.

## 5.7 Ablation study

We conducted two ablation studies. First, we assessed the scalability of our proposed method by varying the number of clients participating in the federated learning process. Second, we applied our method to IID data to compare its performance against the baselines.

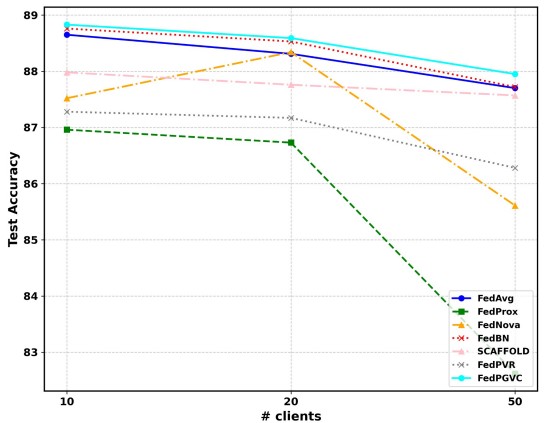

Figure 6: Performance of the proposed model compared to the baselines with varying numbers of clients.

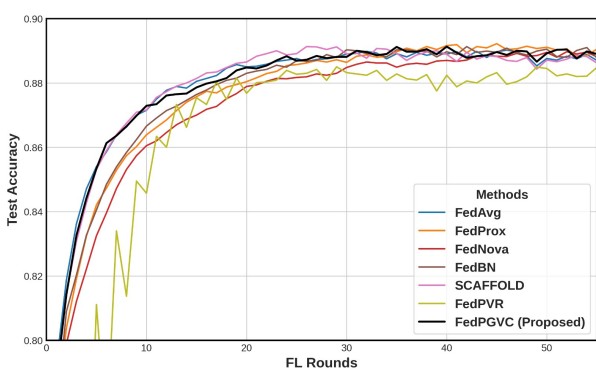

Figure 7: Performance of the proposed model compared to the baselines in FMNIST IID data settings with $\alpha = 100$.

### 5.7.1 Scalibility

To demonstrate the scalability of the proposed FedPGVC in practical settings, we conducted experiments on the FMNIST dataset ($\alpha = 0.5$) with varying numbers of clients: 10, 20, and 50. Figure 6 depicts the performance of the proposed model with the baselines. Notably, when the number of clients increases, the accuracy drop of FedPGVC is considerably low compared to the baselines. This observation highlights the robustness and scalability of our approach, as it can effectively harness a larger pool of clients while maintaining high accuracy.

### 5.7.2 Results on IID data

To assess the efficacy of the proposed FedPGVC method on IID datasets, we created an IID partition of the FMNIST dataset by setting $\alpha = 100$ and compared its performance against other baselines. Fig. 7 shows that FedPGVC performs on par with baseline methods in the IID setting, with error bar graphs in the appendix (Fig. 15(b)). This finding highlights that FedPGVC is not only effective for non-IID data partitions, but also performs well in IID data settings.

### 5.8 Conclusion

This research introduces FedPGVC, an innovative FL-based approach to address the challenges posed by heterogeneous data distributions among clients. By integrating a gradient penalty term into the partial variance control strategy, FedPGVC effectively mitigates the adverse effects of data heterogeneity in federated learning environments. Extensive experiments on diverse datasets reveal FedPGVC's advantage over state-of-the-art baseline methods. Moreover, FedPGVC exhibits faster convergence rates and excellent scalability, consistently delivering performance benefits as the number of clients increases, thus positioning it as a promising solution for large-scale, real-world FL-based computer vision applications.

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

## Appendix

## 6 Convergence Proof

Based on the assumptions in Section 4, we analyze the convergence rate of the proposed FedPGVC method for both convex and non-convex cases.

### 6.0.1 Proof of Theorem 1

**Convex Setting:** For the convex setting, we can derive the following descent lemma. Let us denote the global model parameters at the beginning of round $r$ as $x^{(r)}$, and the local model parameters on client $i$ after $\phi$ local updates as $y_i^{(r,\phi)}$. The modified SGD update rule for the local model on client $i$ is given in Eq. 20:

$$y_i^{(r,\phi+1)} = y_i^{(r,\phi)} - \eta_l \left( g_i^{(r,\phi)} + \rho_i^{(r,\phi)} \odot e \right), \tag{20}$$

where $g_i^{(r,\phi)} = \nabla f_i(y_i^{(r,\phi)}; \xi_i^{(r,\phi)})$ is the stochastic gradient on client $i$, $\rho_i^{(r,\phi)} = \mathbb{E}[g_i^{(r,\phi)} \odot (g_i^{(r,\phi)} - (e \odot x^{(r)}))]$ is a vector denoting the gradient penalty term, $e$ is the masking vector specifying which layers to apply modified SGD on, and $\odot$ denotes element-wise multiplication. The update rule for the global model after aggregating the client models is given in Eq. 21:

$$x^{(r+1)} = \frac{1}{K} \sum_{i=1}^{K} y_i^{(r,E)}, \tag{21}$$

where $E$ is the total number of local updates performed by each client. We will now derive a descent lemma that relates the expected decrease in the objective function after one round of client updates and global model aggregation. Let $F(x) = \frac{1}{K} \sum_{i=1}^{K} f_i(x)$ denote the global objective function, representing the average of all local client objective functions. Since each local objective function $f_i$ is convex, the global objective $F(x)$ is also convex, as described in Eq. 22.

$$f_i(\lambda x + (1-\lambda)y) \leq \lambda f_i(x) + (1-\lambda)f_i(y). \tag{22}$$

By the definition of the global objective $F(x)$, we have:

$$F(x) = \frac{1}{K} \sum_{i=1}^{K} f_i(x).$$

Using the assumption 1 defined in Section 4 i.e., the $L$-smoothness of $F$, we can write:

$$F(x^{(r+1)}) \leq F\left(\frac{1}{K}\sum_{i=1}^{K} y_i^{(r,E)}\right) + \frac{L}{2K}\sum_{i=1}^{K}\|y_i^{(r,E)} - x^{(r)}\|^2.$$

Taking the expectation on both sides, we have:

$$\mathbb{E}[F(x^{(r+1)})] \leq \mathbb{E}\left[F\left(\frac{1}{K}\sum_{i=1}^{K} y_i^{(r,E)}\right)\right] + \frac{L}{2K}\sum_{i=1}^{K}\mathbb{E}\left[\|y_i^{(r,E)} - x^{(r)}\|^2\right].$$

Since $f_i$ is convex, we can apply Jensen's inequality (Eq. 23) on $F$ and obtain Eq 24:

$$\mathbb{E}\left[F\left(\frac{1}{K}\sum_{i=1}^{K} y_i^{(r,E)}\right)\right] \leq \frac{1}{K}\sum_{i=1}^{K}\mathbb{E}[f_i(y_i^{(r,E)})]. \tag{23}$$

$$\mathbb{E}[F(x^{(r+1)})] \leq \frac{1}{K}\sum_{i=1}^{K}\mathbb{E}\left[f_i(y_i^{(r,E)})\right] + \frac{L}{2K}\sum_{i=1}^{K}\mathbb{E}\left[\|y_i^{(r,E)} - x^{(r)}\|^2\right]. \tag{24}$$

We now expand the term $\mathbb{E}[f_i(y_i^{(r,E)})]$ using the sequence of local updates from $\phi = 0$ to $E - 1$ and obtain:

$$\mathbb{E}[f_i(y_i^{(r,E)})] = \mathbb{E}[f_i(y_i^{(r,0)})] + \sum_{\phi=0}^{E-1}\mathbb{E}\left[f_i(y_i^{(r,\phi+1)}) - f_i(y_i^{(r,\phi)})\right].$$

Summing this for all clients, we obtain:

$$\frac{1}{K}\sum_{i=1}^{K}\mathbb{E}[f_i(y_i^{(r,E)})] = \frac{1}{K}\sum_{i=1}^{K}\mathbb{E}[f_i(y_i^{(r,0)})] + \frac{1}{K}\sum_{i=1}^{K}\sum_{\phi=0}^{E-1}\mathbb{E}\left[f_i(y_i^{(r,\phi+1)}) - f_i(y_i^{(r,\phi)})\right].$$

Thus, the above inequality becomes:

$$\mathbb{E}[F(x^{(r+1)})] \leq \frac{1}{K}\sum_{i=1}^{K}\mathbb{E}[f_i(y_i^{(r,0)})] + \frac{1}{K}\sum_{i=1}^{K}\sum_{\phi=0}^{E-1}\mathbb{E}\left[f_i(y_i^{(r,\phi+1)}) - f_i(y_i^{(r,\phi)})\right] + \frac{L}{2K}\sum_{i=1}^{K}\mathbb{E}\left[\|y_i^{(r,E)} - x^{(r)}\|^2\right] \tag{25}$$

Using the assumption 1 again and the update rule defined in Eq. 20, we can further bound the second term of Eq. 25 as:

$$\begin{aligned}
\mathbb{E}\left[f_i\left(y_i^{(r,\phi+1)}\right) - f_i\left(y_i^{(r,\phi)}\right)\right] &\leq \left\langle \nabla f_i\left(y_i^{(r,\phi)}\right), \mathbb{E}\left[y_i^{(r,\phi+1)} - y_i^{(r,\phi)}\right]\right\rangle + \\
&\quad \frac{L}{2}\mathbb{E}\left|y_i^{(r,\phi+1)} - y_i^{(r,\phi)}\right|^2 \\
&= -\eta_l\langle \nabla f_i\left(y_i^{(r,\phi)}\right), g_i^{(r,\phi)} + \rho_i^{(r,\phi)}\odot e\rangle + \\
&\quad \frac{L\eta_l^2}{2}\mathbb{E}\left|g_i^{(r,\phi)} + \rho_i^{(r,\phi)}\odot e\right|^2.
\end{aligned} \tag{26}$$

Substituting the bound obtained from Eq. 26 back into the descent lemma obtained in Eq. 25 and rearranging terms, we get Eq. 27:

$$\mathbb{E}[F(x^{(r+1)})] \leq \frac{1}{K} \sum_{i=1}^{K} f_i\left(x^{(r)}\right) - \frac{\eta_l}{K} \sum_{i=1}^{K} \sum_{\phi=0}^{E-1} \mathbb{E}\left[\left\langle \nabla f_i\left(y_i^{(r,\phi)}\right), g_i^{(r,\phi)} \right\rangle\right]$$

$$-\frac{\eta_l}{K} \sum_{i=1}^{K} \sum_{\phi=0}^{E-1} \mathbb{E}\left[\left\langle \nabla f_i\left(y_i^{(r,\phi)}\right), \rho_i^{(r,\phi)} \odot e \right\rangle\right] + \frac{L\eta_l^2}{K} \sum_{i=1}^{K} \sum_{\phi=0}^{E-1} \mathbb{E}\left|g_i^{(r,\phi)} + \rho_i^{(r,\phi)} \odot e\right|^2 \tag{27}$$

$$+\frac{L}{2K} \sum_{i=1}^{K} \mathbb{E}\left|y_i^{(r,E)} - x^{(r)}\right|^2$$

To telescope the descent lemma obtained in Eq. 27 over multiple FL rounds, we will make use of the inequality shown in Eq. 28, where we use the assumption 2 and the data heterogeneity measure presented in Eq. 17.

$$\sum_{i=1}^{K} \sum_{\phi=0}^{E-1} \left\langle \nabla f_i\left(y_i^{(r,\phi)}\right), g_i^{(r,\phi)} - \nabla f_i\left(y_i^{(r,\phi)}\right) \right\rangle \leq \frac{1}{2} \sum_{i=1}^{K} \sum_{\phi=0}^{E-1} \left(\left|\nabla f_i\left(y_i^{(r,\phi)}\right)\right|^2 + \left|g_i^{(r,\phi)}\right|^2\right)$$

$$\leq \frac{EG^2}{2} + \frac{E\hat{\zeta}^2}{2}. \tag{28}$$

Similarly, we can bound the term involving the gradient penalty term $\rho_i^{(r,k)}$ as:

$$\sum_{i=1}^{K} \sum_{\phi=0}^{E-1} \left\langle \nabla f_i\left(y_i^{(r,\phi)}\right), \rho_i^{(r,\phi)} \odot e \right\rangle \leq \frac{1}{2} \sum_{i=1}^{K} \sum_{\phi=0}^{E-1} \left(\left|\nabla f_i\left(y_i^{(r,\phi)}\right)\right|^2 + \left|\rho_i^{(r,\phi)} \odot e\right|^2\right) \leq \frac{E\hat{\zeta}^2}{2} + \frac{E\hat{\zeta}_e^2}{2}, \tag{29}$$

where $\hat{\zeta}_p^2$ is a measure of the heterogeneity of the gradients for the layers where modified SGD is applied, which is defined in Eq. 30.

$$\hat{\zeta}p^2 = \frac{1}{K} \sum_{i=1}^{K} \mathbb{E}\left|\rho_i^{(r,\phi)} \odot e\right|^2. \tag{30}$$

Substituting bounds obtained from Eq. 29 and using Eq. 30 into the descent lemma obtained in Eq. 27 and telescoping over $R$ FL rounds, we get Eq. 31:

$$\frac{1}{R} \sum_{r=0}^{R-1} \mathbb{E}[F(x^{(r)})] - F^* \leq \frac{1}{R\eta_l} \left(\frac{L\eta_l^2}{2} + \frac{L}{2K}\right) \sum_{r=0}^{R-1} \sum_{i=1}^{K} \mathbb{E}\left|y_i^{(r,E)} - x^{(r)}\right|^2$$

$$+ \frac{1}{2\eta_l}\left(\frac{EG^2}{K} + K\hat{\zeta}^2 + K\hat{\zeta}_p^2\right) + \frac{1}{R}\left(F(x^{(0)}) - F^*\right) \tag{31}$$

where $F^*$ is the optimal value of the average objective function $F(x)$.

To optimize the convergence rate bound, we need to choose the learning rates $\eta_l$ and $\eta_g$ (the global learning rate, which we have not explicitly used yet but will be needed for the final convergence rate). We can set $\eta_g = \sqrt{K}$ and $\eta_l = \min\left\{\frac{1}{26E\eta_g L}, \frac{1}{\sqrt{EL}}\right\}$ to balance the terms in the bound. With these choices, and after algebraic simplifications, we obtain the following convergence rate bound for non-convex functions as given in Eq. 32:

$$\frac{1}{R} \sum_{r=0}^{R-1} \mathbb{E}[F(x^{(r)})] - F^* = \mathcal{O}\left(\frac{G\sqrt{K}}{\sqrt{KR}} + \frac{\hat{\zeta}\sqrt{E}}{\sqrt{R}} + \frac{\hat{\zeta}_p\sqrt{E}}{\sqrt{KR}} + \frac{F(x^{(0)}) - F^*}{R}\right). \tag{32}$$

This bound shows that the convergence rate of our approach depends on the number of communication rounds $R$, the number of local updates $E$, the gradient bound $G$, the overall data heterogeneity $\hat{\zeta}$, and the heterogeneity of the gradients for the layers where modified SGD is applied, $\hat{\zeta}_e$. Therefore, the final convergence rate of the algorithm is $\mathcal{O}\left(\frac{1}{\sqrt{R}}\right)$.

**Non-Convex setting:**

By applying the Lipschitz smoothness property of $F(x)$, as defined in Assumption 1(b) of Section 4, at round $r$, we obtain:

$$\mathbb{E}[F(x^{(r+1)})] \leq \mathbb{E}[F(x^{(r)})] + \mathbb{E}\left\langle \nabla F(x^{(r)}), x^{(r+1)} - x^{(r)} \right\rangle + \frac{L}{2}\mathbb{E}\left\| x^{(r+1)} - x^{(r)} \right\|^2,$$

where $x^{(r)}$ is the global model at round $r$, and $x^{(r+1)} = \frac{1}{K}\sum_{i=1}^{K} y_i^{(r,E)}$ is the updated global model after aggregating the local models $y_i^{(r,E)}$ from all clients. Since $x^{(r+1)} = \frac{1}{K}\sum_{i=1}^{K} y_i^{(r,E)}$, the inner product term becomes:

$$\mathbb{E}\left\langle \nabla F(x^{(r)}), x^{(r+1)} - x^{(r)} \right\rangle = \mathbb{E}\left\langle \nabla F(x^{(r)}), \frac{1}{K}\sum_{i=1}^{K}(y_i^{(r,E)} - x^{(r)}) \right\rangle.$$

Taking the expectation over the stochastic gradient updates, we get:

$$\mathbb{E}\left\langle \nabla F(x^{(r)}), x^{(r+1)} - x^{(r)} \right\rangle = -\eta_l E \mathbb{E}\left[\|\nabla F(x^{(r)})\|^2\right] + \frac{L\eta_l^2 E^2 \sigma^2}{2},$$

where $\eta_l$ is the local learning rate and $E$ is the number of local updates on each client. Since the variance of the local updates $y_i^{(r,E)}$ grows linearly with the number of local steps $E$, we can bound the term $\|x^{(r+1)} - x^{(r)}\|^2$ as:

$$\mathbb{E}\left\| x^{(r+1)} - x^{(r)} \right\|^2 \leq \eta_l^2 E^2 \sigma^2.$$

Substituting this bound into the smoothness inequality, we obtain:

$$\mathbb{E}[F(x^{(r+1)})] \leq \mathbb{E}[F(x^{(r)})] - \eta_l E \mathbb{E}\left[\|\nabla F(x^{(r)})\|^2\right] + \frac{L\eta_l^2 E^2 \sigma^2}{2}.$$

Summing the inequality over $R$ communication rounds, we get:

$$\sum_{r=0}^{R-1} \mathbb{E}\left[\|\nabla F(x^{(r)})\|^2\right] \leq \frac{F(x^{(0)}) - F(x^*)}{\eta_l E} + \frac{L\eta_l E \sigma^2 R}{2},$$

where $F(x^*)$ is the optimal value of the global objective. Dividing both sides by $R$ yields Eq. 33.

$$\frac{1}{R}\sum_{r=0}^{R-1} \mathbb{E}\left[\|\nabla F(x^{(r)})\|^2\right] \leq \frac{F(x^{(0)}) - F(x^*)}{\eta_l E R} + \frac{L\eta_l E \sigma^2}{2} \tag{33}$$

To minimize the right-hand side of the above inequality (Eq. 33), the local learning rate $\eta_l$ is selected such that the two terms on the right-hand side balance. So we set $\eta_l = \Theta\left(\frac{1}{\sqrt{ER}}\right)$ and get the final convergence rate as below:

$$\frac{1}{R}\sum_{r=0}^{R-1} \mathbb{E}\left[\|\nabla F(x^{(r)})\|^2\right] = O\left(\frac{1}{\sqrt{ER}}\right) + O\left(\frac{\sigma^2}{\sqrt{ER}}\right).$$

So in the non-convex setting, the proposed FedPGVC approach achieves a convergence rate of $O\left(\frac{1}{\sqrt{ER}}\right)$, where $E$ is the number of local updates and $R$ is the number of communication rounds. This result shows that increasing the number of local updates $E$ improves the convergence rate, but too large a learning rate $\eta_l$ or too many local updates may introduce additional variance due to stochastic gradients.

# 7 Applying PGVC on the different layers of the model

To evaluate the impact of incorporating gradient variance control in various neural network layers, we conducted experiments on the FMNIST dataset with $\alpha = 0.5$. As illustrated in Fig. 8, our findings reveal that applying variance reduction in the final layers accelerates convergence and achieves the highest top-1 accuracy. Given that the proposed approach partially applies the gradient variance control technique in the last layers of the neural network, we investigated the effects of incorporating variance reduction in different layers. We conducted experiments on the FMNIST dataset with $\alpha = 0.5$, as shown in Fig. 8. The results indicate that initiating variance reduction in the final layers of the model facilitates faster convergence and achieves the highest top-1 accuracy.

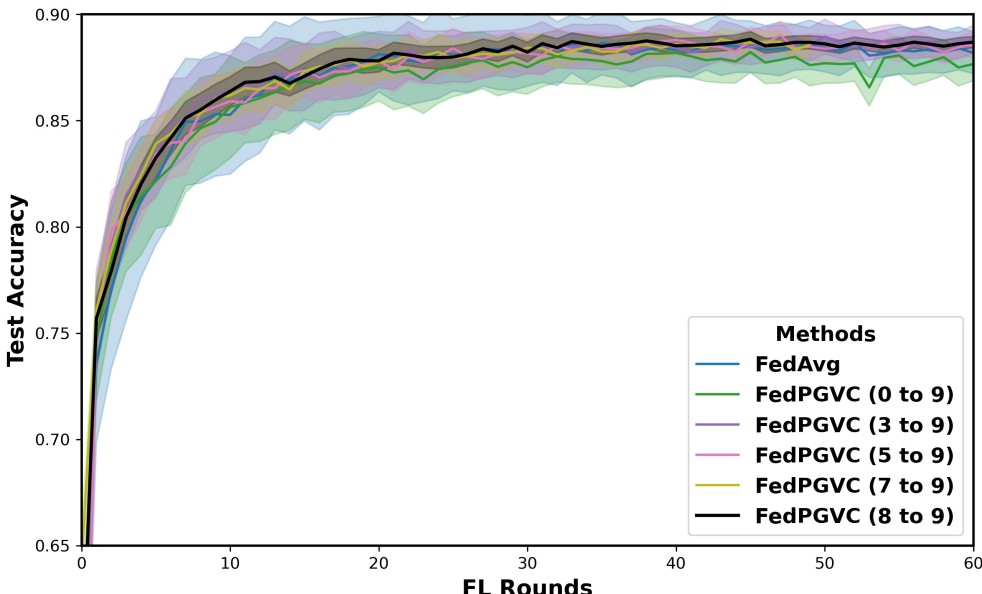

Figure 8: Performance of applying partial gradient variance control on different layers of the CNN model on the FMNIST dataset with $\alpha = 0.5$.

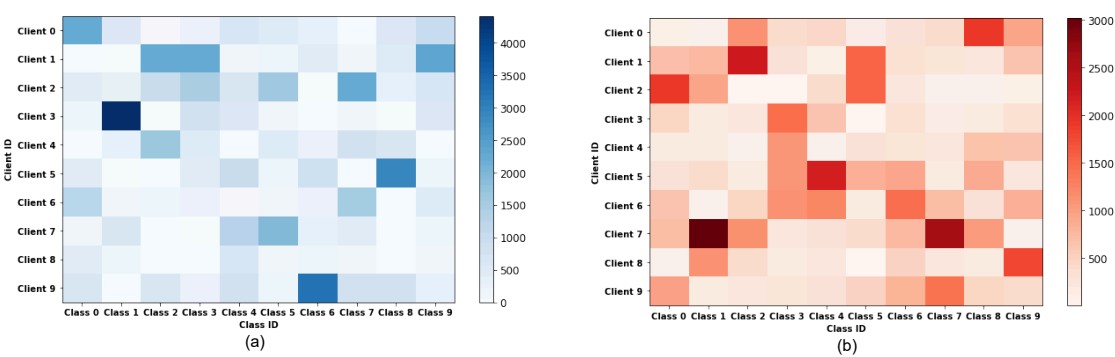

Figure 9: Distribution of MNIST data, indicating the number of images per client per class, varied according to different levels of heterogeneity, with (a) $\alpha = 0.5$ representing severe non-IID and (b) $\alpha = 1.0$ indicating moderate non-IID.

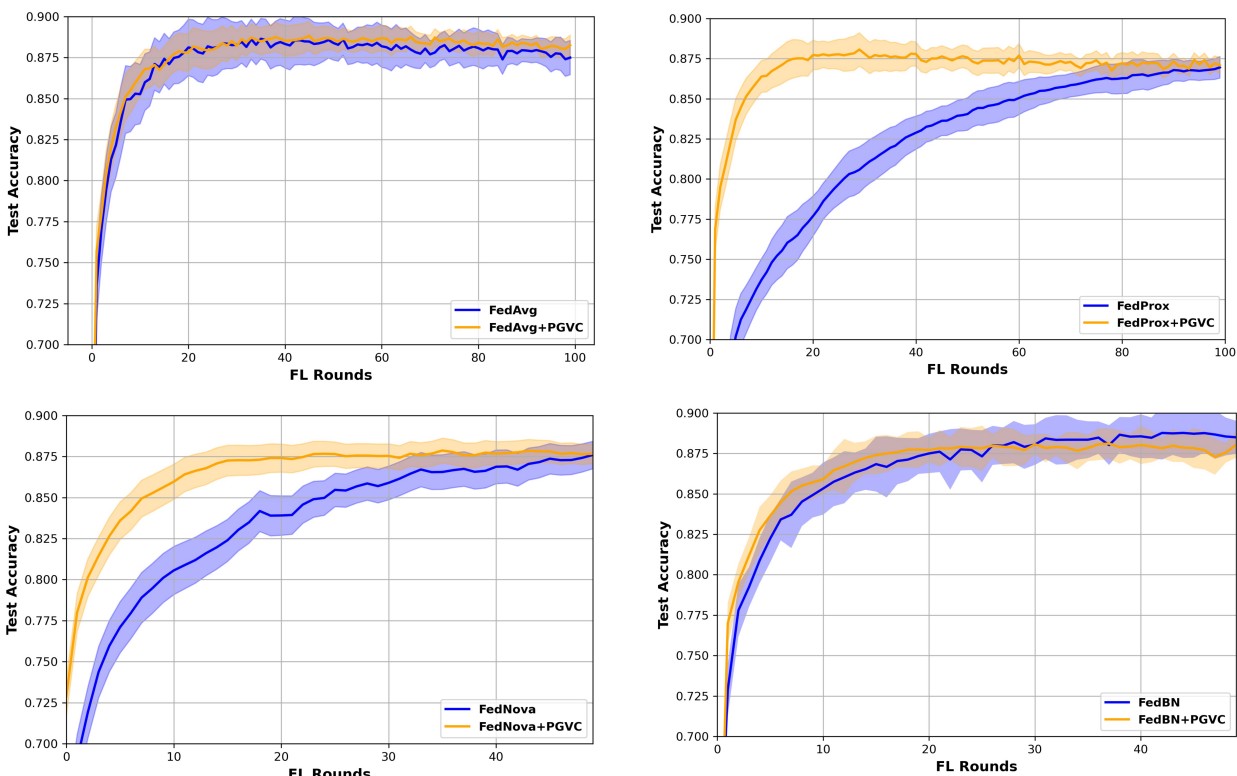

Figure 10: Learning curves with error bars illustrating the integration of the proposed PGVC technique with the existing algorithms on the FMNIST dataset at $\alpha = 0.5$.

Table 6: Extended results, including standard deviations, showing the effect of applying the proposed PGVC method to existing popular baselines on the FMNIST and CIFAR100 datasets with $\alpha = 0.5$.

| Method | FMNIST | CIFAR100 |
|---|---|---|
| FedAvg | $88.65 \pm 0.06$ | $24.25 \pm 0.22$ |
| FedAvg + PGVC | $\mathbf{88.83 \pm 0.11}$ | $\mathbf{25.29 \pm 0.20}$ |
| FedProx | $86.96 \pm 0.22$ | $\mathbf{24.89 \pm 0.17}$ |
| FedProx + PGVC | $\mathbf{88.07 \pm 0.17}$ | $24.58 \pm 0.11$ |
| FedBN | $\mathbf{88.86 \pm 0.12}$ | $\mathbf{25.12 \pm 0.18}$ |
| FedBN + PGVC | $88.11 \pm 0.15$ | $23.86 \pm 0.14$ |
| FedNova | $87.52 \pm 0.11$ | $22.29 \pm 0.16$ |
| FedNova + PGVC | $\mathbf{87.87 \pm 0.14}$ | $\mathbf{24.82 \pm 0.19}$ |

## 8 Statistical Test

To evaluate the statistical significance of performance differences between the baseline and the proposed method, we conducted McNemar's test (McNemar, 1947). Unlike overall test accuracy, which quantifies correctness at an aggregate level, McNemar's test focuses on pairwise prediction differences, making it particularly sensitive to variations in error distributions. We performed the test at a 95% confidence level, and the results are presented in Tables 7 and Table 8. In all the cases, the computed p-values were below 0.05, allowing us to reject the null hypothesis. This confirms a statistically significant difference between the baseline and our proposed method.

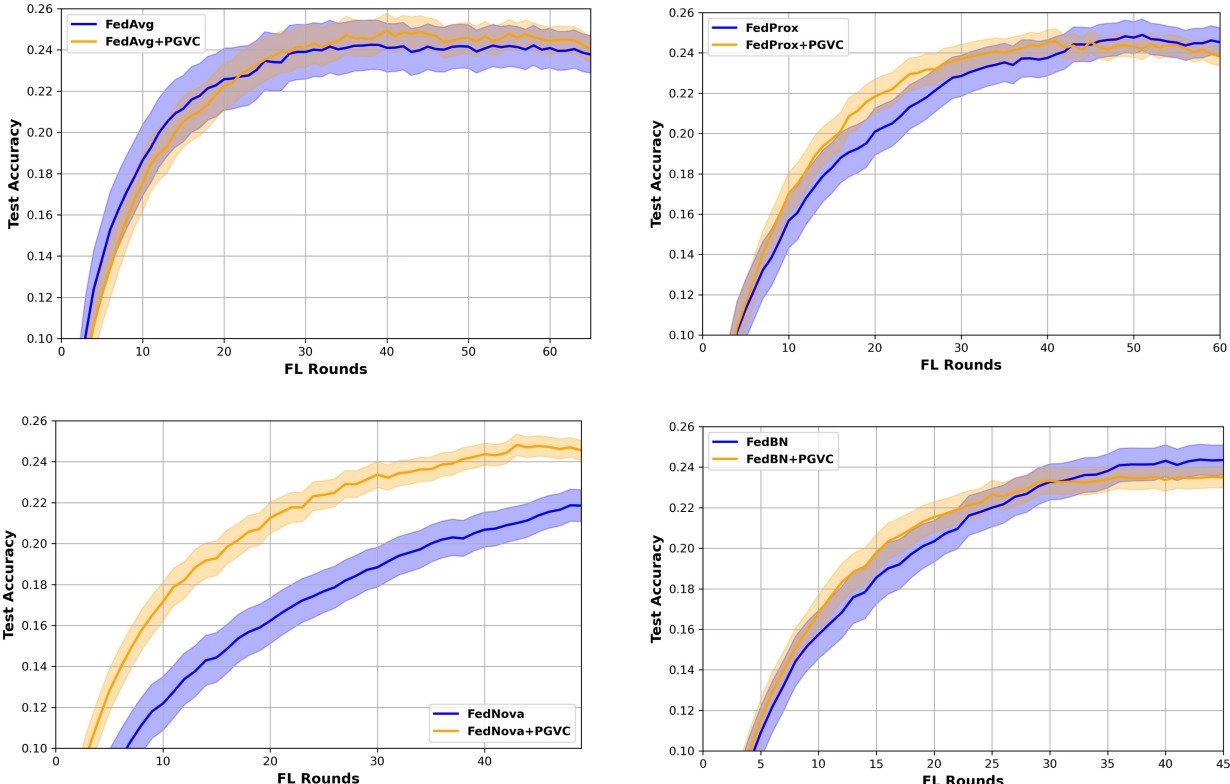

Figure 11: Learning curves with error bars illustrating the integration of the proposed PGVC technique with the existing algorithms on the CIFAR100 dataset at $\alpha = 0.5$.

## 9    Limitation

While the proposed approach demonstrates significant improvements across various datasets, it is essential to acknowledge certain limitations. Although the method effectively reduces gradient variability in the final layers, the computation of the gradient penalty term may introduce additional processing overhead on the client side. This increased computational cost could be a constraint for resource-limited devices, even though the method is designed to minimize communication overhead. To address this challenge, future research could focus on reducing the computational complexity of the gradient penalty term without sacrificing the method's effectiveness. Approaches like model pruning or quantization may be explored to make the method more viable for devices with limited resources. Additionally, incorporating differential privacy techniques into the FedPGVC framework could broaden its applicability in privacy-sensitive areas, such as healthcare or finance. Investigating the interaction between differential privacy and the gradient penalty term, as well as its impact on model performance, would be a valuable direction for future research.

Table 7: p-values indicating the statistical significance of test accuracies on MNIST, FMNIST, and CIFAR100 under varying data heterogeneity, relative to the Proposed method.

|  | MNIST | | FMNIST | | CIFAR100 | |
| --- | --- | --- | --- | --- | --- | --- |
|  | $\alpha = 0.5$ | $\alpha = 1.0$ | $\alpha = 0.5$ | $\alpha = 1.0$ | $\alpha = 0.5$ | $\alpha = 1.0$ |
| FedAvg | 3.21e-12 | 7.45e-9 | 2.12e-8 | 5.33e-7 | 8.54e-15 | 9.18e-11 |
| FedProx | 6.78e-10 | 4.23e-8 | 1.29e-7 | 2.77e-6 | 2.61e-6 | 5.44e-9 |
| FedNova | 5.21e-9 | 1.14e-6 | 3.87e-6 | 6.45e-5 | 4.73e-8 | 1.07e-9 |
| FedBN | 4.12e-8 | 9.67e-7 | 7.32e-6 | 8.89e-5 | 3.21e-8 | 5.56e-10 |
| SCAFFOLD | 1.78e-7 | 5.22e-6 | 2.89e-5 | 4.76e-4 | 9.33e-10 | 2.23e-9 |
| FedPVR | 2.34e-6 | 8.73e-5 | 1.11e-4 | 3.12e-3 | 5.67e-7 | 6.88e-6 |

Table 8: p-values indicating significance compared to the Proposed method across various complex models on CIFAR-100, Tiny-ImageNet, and QQP datasets.

|  | CIFAR100 (ResNet18) | CIFAR100 (ViT) | Tiny-ImageNet (ResNet18) | QQP (LSTM) |
|---|---|---|---|---|
| FedAvg | 2.21e-4 | 3.41e-6 | 5.12e-5 | 1.87e-3 |
| FedProx | 5.14e-5 | 2.89e-7 | 3.78e-4 | 6.92e-4 |
| FedNova | 1.33e-6 | 4.57e-9 | 1.02e-6 | 3.10e-5 |
| FedBN | 3.87e-4 | 1.12e-6 | 8.45e-5 | 2.43e-3 |
| SCAFFOLD | 4.75e-4 | 2.65e-5 | 7.89e-6 | 3.20e-4 |
| FedPVR | 9.56e-8 | 7.22e-11 | 1.31e-9 | 9.87e-6 |

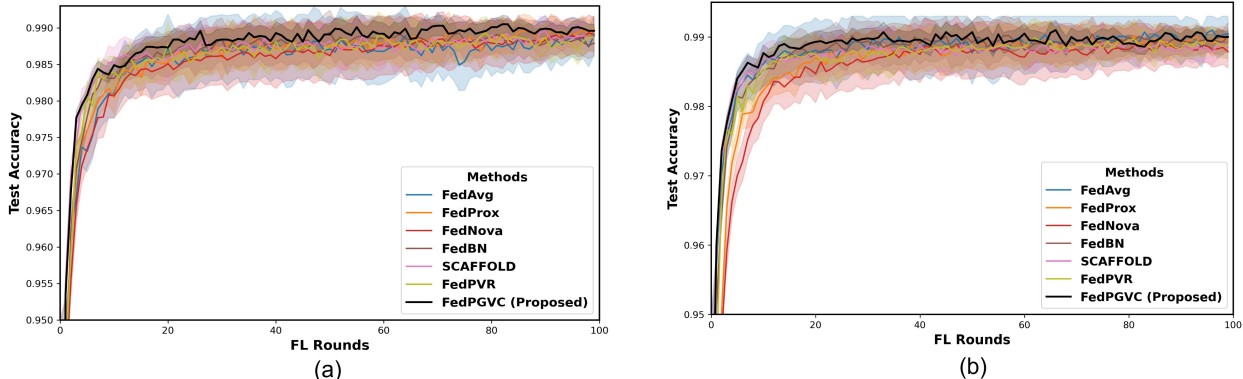

Figure 12: Performance comparison of proposed FedPGVC with baseline approaches with error bars: (a) and (b) depict the graphs on the MNIST dataset for $\alpha = 0.5$ and 1.0 respectively.

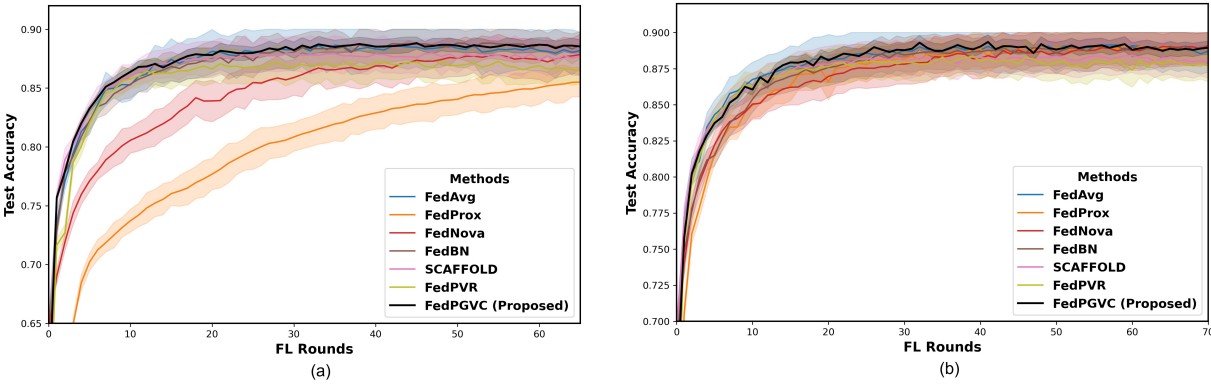

Figure 13: Performance comparison of proposed FedPGVC with baseline approaches with error bars: (a) and (b) depict the graphs on the FMNIST dataset for $\alpha = 0.5$ and 1.0 respectively.

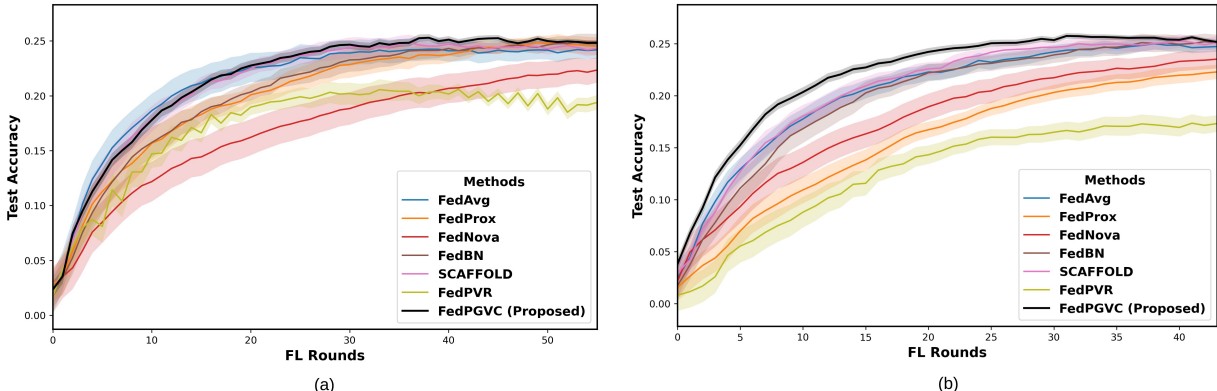

Figure 14: Performance comparison of proposed FedPGVC with baseline approaches with error bars: (a) and (b) depict the graphs on the CIFAR100 dataset for $\alpha = 0.5$ and $1.0$ respectively.

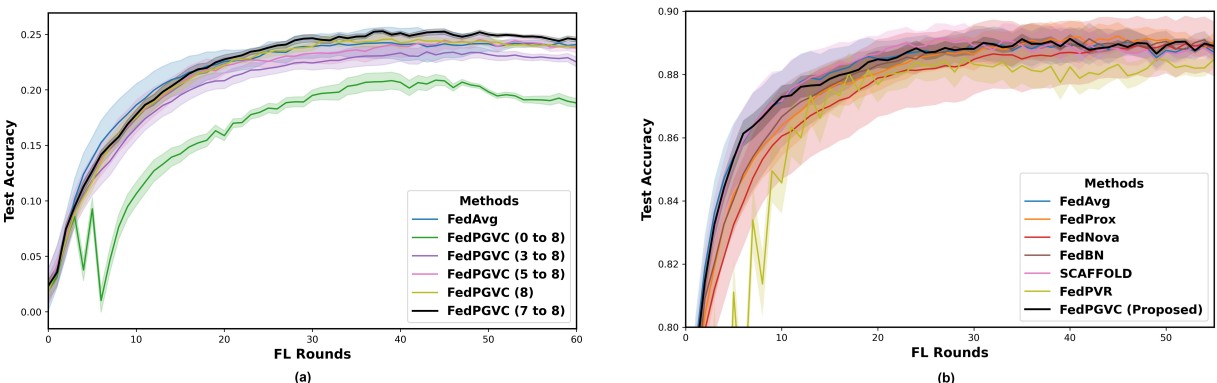

Figure 15: (a) Performance of applying partial gradient variance control on different layers of the CNN model with error bars on the CIFAR100 dataset with $\alpha = 0.5$. (b) Performance of the proposed model compared to the baselines with error bars in FMNIST IID data settings with $\alpha = 100$.

