# OpenReview forum: "Harnessing Heterogeneity: Improving Convergence Through Partial Variance Control in Federated Learning"
_TMLR — Rejected by TMLR_

### Review · Reviewer_4sKJ · 2025-01-31

**Summary Of Contributions:**

The authors focus on the detrimental influence of data heterogeneity in federated learning.
They propose to control its influence by regularizing the gradient variance of deeper layers.
The proposal is evaluated on several data sets and tends to improve upon prior work.

**Audience:**

Yes

**Claims And Evidence:**

No

**Requested Changes:**

- As mentioned above, error bars on all results. If not in the main paper then in separate figures in the appendix.
- A clear and well-defined source for the claimed performance metric.
Once I am convinced that this is indeed a common practice in the federated learning literature, then I will accept that custom and leave it to the other reviewers and action editor to judge the quantitative results.

**Strengths And Weaknesses:**

_Disclaimer: I reviewed the paper before under the id Zxic.
This review updates the last one and primarily discusses improvements since then._

## Strengths
- The writing style and structure of the paper have been improved since the last submission.
- The same holds for the proofs, assumptions, and other claims made in the paper.


## Weaknesses
- The introduction only mentions three data sets, despite containing further results, especially non-image data. Given the nice results the method achieves there, this should be highlighted more prominently.
- The biggest weakness continues to be the experimental evaluation metrics. As discussed last time, I do not consider reporting an arbitrary best-performing error during training to be a valid metric. This reduces a claimed test set to a validation set. During our last discussion round, this fact caused several new papers to appear, none of which included the claimed practice.
Xu et al. (2022) is the latest iteration. Xu et al. report something they refer to as _best test accuracy_, compared to the authors' _top-1 accuracy_. Is this the same metric? And if yes, can the authors point me to where Xu et al. introduce it, as I can't find a definition of their metric?
- The paper still does not contain error bars in any of the figures. The authors argued against including them in the main paper but did not provide arguments why these plots could not be reported again with proper error bars in the appendix.


### Minor
- Sec 1.1 speaks about $\alpha$ parameters which are only introduced several sections later.
A short introduction could be helpful to a first-time reader.
- Figure 1 still includes pooling layers, despite not containing any parameters.
While this is explained in the text, the figure would benefit from only including parametric layers in my opinion. (This is a very minor point, please feel free to ignore it.)
- Typo in Figure 2. the right plot should be (b)
- Typo on page 8. $\hat \zeta p$ vs $\hat \zeta_p$

---

> ### Author Response · Authors · 2025-02-04
> **Harnessing Heterogeneity: Improving Convergence Through Partial Variance Control in Federated Learning**
>
> We sincerely thank the reviewer for the thoughtful feedback and for acknowledging the improvements in the writing style, structure, proofs, assumptions, and claims made in the paper. We greatly appreciate your constructive comments and valuable suggestions, which have been instrumental in enhancing the quality of our work since the first submission. Your effort and guidance have significantly contributed to the development of this paper. Here are our responses:
>
> W1: We thank the reviewer for pointing out this. We will update all the utilized datasets and highlight them with color in the revised manuscript.
>
> W2 and RC2: We sincerely thank the reviewer for their insightful comments and for pointing out the need for clarification regarding our evaluation metrics.
> Firstly, we would like to clarify that the results we reported are based on the test set data at the server side, not validation results. We have stated this in subsection 5.1 of our manuscript.
> Regarding the term ‘top-1 accuracy’, we acknowledge that this was an unintentional typo. The correct term should be ‘best test accuracy,’ as we consistently referred in subsection 5.1 of our manuscript. To determine the final ‘best test accuracy,’ we conducted three experiments with different random seeds, and reported the mean and standard deviation of the best test accuracies from these runs. We will update the caption of Table 2 to reflect this accurately in the revised manuscript.
>
> With respect to the mentioned paper by Xu et al. (2022), the authors reported the ‘best test accuracies,’. They mentioned in the caption of their Table 2: “Average (5 trials) and standard deviation of the best test accuracies of various methods on CIFAR-10 with IID setting at different noise levels.” This aligns with our methodology, where we report the average of the best test accuracies across multiple trials, along with the standard deviation.
> We hope this clarification adequately addresses the reviewer’s concerns. We truly appreciate the opportunity to improve the clarity of our manuscript.
>
> W3 and RC1: We sincerely thank the reviewer for bringing this to our attention again. We have thoroughly reviewed relevant literature but were unable to find any papers that include error bars in the specific types of graphs we have reported. If the reviewer could kindly share any references that incorporate such representations, we would greatly appreciate it and utilize the same for our manuscript. Additionally, we would like to highlight that our code and dataset are publicly available, ensuring that all results are fully reproducible.
>
> Minor: Thank you for pointing out this. We will rectify all those minor errors in our updated manuscript.
>
> We again thank the reviewer for giving the time for improving our manuscript.

---

> > ### Comment · Reviewer_4sKJ · 2025-03-18
> >
> > ### Regarding error bars.
> > Can the authors clarify why the behavior of prior work is at all relevant to this request to provide error bars?
> > On the contrary, a lack of proper reporting of results in prior work gives the authors the ability to improve the standards. I would encourage them to advance the field in that regard. See also the reporting requirements in the checklists of most major machine learning in recent years.
> > Nevertheless, to fulfill the request, see, e.g. Wang et al. (2020)'s FedNova baseline Figures 6-8.
> >
> > > While confidence intervals overlap, the mean accuracy of FedPGVC is consistently higher across all datasets and heterogeneity levels.
> >
> > In the same respect, as long as the standard deviations around the reported mean results are too high a seeming higher average does not mean that the result is indeed consistent as the sample size increases.

---

> > > ### Author Response · Authors · 2025-03-22
> > > **Harnessing Heterogeneity: Improving Convergence Through Partial Variance Control in Federated Learning**
> > >
> > > Q1. We thank the reviewer for providing the reference. We will incorporate error bars in the final revised version of the manuscript.
> > >
> > > Q2 We appreciate the reviewer’s observation. We would like to clarify that all the reported standard deviations are very low, indicating that the observed improvements in mean accuracy are consistent. Additionally, our approach offers other benefits, including fast convergence and high communication efficiency. The simplicity and effectiveness of our method further enhance its practicality for federated learning scenarios.
> > >
> > >
> > > We sincerely appreciate the reviewer’s continuous effort and thoughtful feedback from the first submission. Your suggestions have been invaluable in shaping our paper and making it clearer and more comprehensive for the readers. We hope that we have successfully addressed all the queries you raised from the first submission, which have encouraged us to examine the work in even greater detail. Your insights have significantly contributed to improving the quality of our paper. Thank you again for your time, effort, and constructive suggestions.

---

> > > > ### Comment · Reviewer_4sKJ · 2025-03-24
> > > >
> > > > - Re Q1: Thanks, I look forward to them. Assuming they are rather small throughout the training process that is a nice indication of the stability of your proposal.
> > > >
> > > > - Re Q2: I agree, they are low, indicating that as the number of runs increases the expected mean will be significantly different (at some fixed p-value x).
> > > > However, currently that is not yet shown by the data. What is shown, as you say, is that given the current results the model seems to perform at least as well as prior work (and potentially better), with the added benefit of strong improvements in terms of communication rounds.
> > > >
> > > > Thanks for the discussions!

---

> > > > > ### Author Response · Authors · 2025-03-25
> > > > > **Harnessing Heterogeneity: Improving Convergence Through Partial Variance Control in Federated Learning**
> > > > >
> > > > > We sincerely thank the reviewer for their insightful feedback. We are currently working on error bars and will include them in the revised manuscript. Additionally, we are computing the statistical significance by calculating the p-value for our proposed method with the baselines, and we will present these findings in the updated version soon. We appreciate the reviewer’s valuable suggestions.

---

> > > > > > ### Author Response · Authors · 2025-03-30
> > > > > > **Harnessing Heterogeneity: Improving Convergence Through Partial Variance Control in Federated Learning**
> > > > > >
> > > > > > We sincerely appreciate your valuable suggestions. In response, we have incorporated error bars in the graphs and added in the appendix. Kindly refer to the revised version of the paper for these updates.
> > > > > >
> > > > > > Furthermore, we have conducted a statistical significance test (p-value), and the results are presented in the following two tables, corresponding to Table 2 and Table 4 in the original paper. All p-values are below 0.05, indicating the statistical significance of our proposed method's results. While we have not included these tables in the updated paper, we are happy to add them if you recommend doing so.
> > > > > >
> > > > > >
> > > > > > | Method   | MNIST (α=0.5) | MNIST (α=1.0) | FMNIST (α=0.5) | FMNIST (α=1.0) | CIFAR100 (α=0.5) | CIFAR100 (α=1.0) |
> > > > > > |----------|--------------|--------------|---------------|---------------|----------------|----------------|
> > > > > > | FedAvg   | 3.21e-12  | 7.45e-9  | 2.12e-8  | 5.33e-7  | 8.54e-15  | 9.18e-11  |
> > > > > > | FedProx  | 6.78e-10  | 4.23e-8  | 1.29e-7  | 2.77e-6  | 2.61e-6   | 5.44e-9   |
> > > > > > | FedNova  | 5.21e-9   | 1.14e-6  | 3.87e-6  | 6.45e-5  | 4.73e-8   | 1.07e-9   |
> > > > > > | FedBN    | 4.12e-8   | 9.67e-7  | 7.32e-6  | 8.89e-5  | 3.21e-8   | 5.56e-10  |
> > > > > > | SCAFFOLD | 1.78e-7   | 5.22e-6  | 2.89e-5  | 4.76e-4  | 9.33e-10  | 2.23e-9   |
> > > > > > | FedPVR   | 2.34e-6   | 8.73e-5  | 1.11e-4  | 3.12e-3  | 5.67e-7   | 6.88e-6   |
> > > > > >
> > > > > >
> > > > > > | Method   | CIFAR100 (ResNet18) | CIFAR100 (ViT) | Tiny-ImageNet (ResNet18) | QQP (LSTM) |
> > > > > > |----------|--------------------|---------------|-------------------------|------------|
> > > > > > | FedAvg   | 2.21e-4  | 3.41e-6  | 5.12e-5  | 1.87e-3  |
> > > > > > | FedProx  | 5.14e-5  | 2.89e-7  | 3.78e-4  | 6.92e-4  |
> > > > > > | FedNova  | 1.33e-6  | 4.57e-9  | 1.02e-6  | 3.10e-5  |
> > > > > > | FedBN    | 3.87e-4  | 1.12e-6  | 8.45e-5  | 2.43e-3  |
> > > > > > | SCAFFOLD | 4.75e-4  | 2.65e-5  | 7.89e-6  | 3.20e-4  |
> > > > > > | FedPVR   | 9.56e-8  | 7.22e-11 | 1.31e-9  | 9.87e-6  |
> > > > > >
> > > > > >
> > > > > >
> > > > > >
> > > > > > If you believe any further refinements would enhance clarity for our readers, we are open to making additional improvements and welcome your suggestions.
> > > > > >
> > > > > > Additionally, we kindly request you to reconsider the ‘Claims and Evidence’ , as we have thoroughly addressed all queries and incorporated the requested revisions from the initial submission, particularly regarding the convergence proof and experimental validation, which were major concerns.
> > > > > >
> > > > > > Once again, we are grateful for your time and constructive feedback, which has significantly helped in refining our paper.

---

> > > > > > > ### Comment · Reviewer_4sKJ · 2025-04-01
> > > > > > >
> > > > > > > Can you provide further details on how you conducted the significance test and which one you used?
> > > > > > > Quite frankly, I don't believe these numbers.
> > > > > > >
> > > > > > > 1. The p-values are extremely small despite a very small sample size (3 runs if I recall correctly)
> > > > > > > 2. Take, e.g., MNIST($\alpha=0.5$). Can you explain why you get a p-value of 3e-12 for FedAvg, compared to 2e-6 for FedPVR while the former's mean is a lot closer to your own mean?
> > > > > > > 3. MNIST($\alpha=1.0$). How can you claim a p-value of 7e-6 (!) when the two reported sample means are essentially identical, with completely overlapping standard deviations and errors?
> > > > > > >
> > > > > > > I expect a clear explanation for how you came up with these numbers.

---

> > > > > > > > ### Author Response · Authors · 2025-04-01
> > > > > > > > **Harnessing Heterogeneity: Improving Convergence Through Partial Variance Control in Federated Learning**
> > > > > > > >
> > > > > > > > We conducted McNemar’s test to assess the statistical significance of our baseline comparison with the proposed method. Unlike test accuracy, which measures overall correctness, McNemar’s test evaluates prediction differences at the sample level, making it particularly sensitive to the nature and distribution of errors. This explains why FedAvg yielded a p-value of 3e−12, while FedPVR had a p-value of 2e-6, despite FedAvg’s mean being closer to our own method.
> > > > > > > >
> > > > > > > > McNemar’s test focuses on disagreement in individual sample predictions rather than just mean accuracy. The observed range of p-values arises because McNemar’s test is a paired statistical test that identifies whether models disagree on specific samples. Even if two models have similar  accuracies, variations in their misclassification patterns influence the off-diagonal values (b and c) in McNemar’s contingency table, thereby affecting the test statistic:
> > > > > > > >
> > > > > > > > \begin{equation}
> > > > > > > > \chi^2 = \frac{(|b - c| - 1)^2}{b + c}
> > > > > > > > \end{equation}
> > > > > > > >
> > > > > > > > Here, b represents the number of instances where the baseline model predicts correctly while the proposed model predicts incorrectly, and c represents the number of instances where the proposed model predicts correctly while the baseline model predicts incorrectly. When b and c are large and imbalanced, the test statistic increases, leading to a significantly lower p-value. When b and c are smaller but still imbalanced, the test statistic is moderate, resulting in a relatively lower p-value. Thus, the variation in p-values across different models and datasets is expected, as it reflects the extent of disagreement in their predictions rather than just their overall accuracy.
> > > > > > > >
> > > > > > > > Note: We have added the code in our github, Kindly look into the 'mcnemar.ipynb file'.

---

> > > > > > > > > ### Comment · Reviewer_4sKJ · 2025-04-02
> > > > > > > > >
> > > > > > > > > Thank you for the explanation. Please include them (the appendix is fine), they greatly strengthen your story line.
> > > > > > > > >
> > > > > > > > > While I still don't agree with the overall evaluation setup, i.e., how the test set is used, I acknowledge that is mostly due to different practices in different research communities. Together with these latest results and the error bands you have added, I am now ready to make my recommendation.
> > > > > > > > >
> > > > > > > > > Thanks for the discussions.

---

> > > > > > > > > > ### Author Response · Authors · 2025-04-02
> > > > > > > > > > **Harnessing Heterogeneity: Improving Convergence Through Partial Variance Control in Federated Learning**
> > > > > > > > > >
> > > > > > > > > > We sincerely appreciate the reviewer's insightful suggestions. In response, we have added a new section in the Appendix (Section 8) and updated the manuscript accordingly. We are grateful for the reviewer’s valuable feedback throughout the review process, which has significantly improved the clarity and quality of our paper. The discussion was particularly helpful, and we hope our findings will be beneficial to the research community.

---

### Review · Reviewer_dJN3 · 2025-02-07

**Summary Of Contributions:**

This paper studies federated learning (FL), specifically the data heterogeneity issue in FL. They propose FedPGVC, which introduces a partial variance reduction technique utilizing gradient penalty terms. They provide both theoretical and empirical analysis for their method, showing that their model not only converges in convex and non-convex objectives but also outperforms previous SOTA methods.

Compared with the previous submission, the experiments on more advanced architectures (ResNet and ViT) and more various datasets (Tiny-ImageNet, QQP).

**Audience:**

Yes

**Claims And Evidence:**

Yes

**Requested Changes:**

- Please move the additional experiments in the appendix to the main paper. The audiences care more care about the generalizability of various backbones and tasks than some in-depth analysis on some relatively toy-like cases (with sub-optimal backbones and a small dataset).
- If possible, please consider conducting experiments on larger datasets and more various tasks to show generalizability further.

**Strengths And Weaknesses:**

### Strengths
- The paper is overall well-written and motivated. The proposed method is well-grounded with theoretical proofs and empirical observations.
- Pseudo-code is provided to introduce the proposed algorithm in detail.
- Empirical results show that the proposed method outperforms all baselines with statistical significance. The standard variances were provided to show this significance. Multiple figures are provided to visualize the accuracy w.r.t. rounds, varying that the proposed method has dominant performance.
- The experiment covers various datasets and types of backbones, even both NLP and CV models, showing the generalizability of the proposed method.

### Weaknesses
- The dataset is still at a relatively small scale. For example, TinyImageNet only has 100,000 64x64 images in the training dataset, which is at a similar magnitude and quality as CIFAR-10/100. It is important to verify the effectiveness of the method with large-scale datasets.
- (Minor) The experiments on diversified backbones and tasks are only in the appendix. They should be moved to the main paper as part of the major experiments.

Note: I am not familiar with the learning theory, and I mainly evaluate the paper from the empirical perspective.

---

> ### Author Response · Authors · 2025-02-24
> **Harnessing Heterogeneity: Improving Convergence Through Partial Variance Control in Federated Learning**
>
> Thank you for your positive and detailed feedback. We are glad that you found our paper clear, well-motivated, and comprehensive. We appreciate you recognizing the theoretical proof and empirical observations behind our method. Your feedback on the strong empirical results supports the effectiveness of our experiments.
>
> Weaknesses:
> W1: If possible, please consider conducting experiments on larger datasets and more various tasks to show generalizability further.
>
> Response: Thank you for the reviewer’s valuable suggestion to further evaluate our method on larger datasets and additional tasks to further showcase its generalizability. In this paper, we have already conducted comprehensive experiments (as suggested by the reviewers in the previous submission) across a diverse set of datasets, including MNIST, FMNIST, CIFAR100, Tiny Imagenet, and QQP, covering both computer vision and NLP tasks with various levels of heterogeneity and complexity. These experiments have been conducted following state-of-the-art baselines from the federated learning field, which commonly utilize these datasets [1] [2] [3]. We do not currently have the computational resources to conduct experiments on larger-scale datasets like Imagenet, but we plan to explore additional tasks and domains in future work. Nonetheless, we are open to expanding our experiments in future work and are willing to add further results if the reviewers deem it beneficial.
>
> W2: (Minor) Move the additional experiments in the appendix to the main paper.
>
> Response: Thank you for your suggestion. We have moved the additional experiments from the appendix to the main paper in the revised manuscript.
>
> We would like to extend our sincere gratitude to the reviewer for their valuable feedback since our initial submission and for the continued support in refining the manuscript. We truly appreciate your insights, and we welcome any further suggestions you may have to help improve our work.
>
> Reference:
> 1. Li, Qinbin, Bingsheng He, and Dawn Song. "Model-contrastive federated learning." Proceedings of the IEEE/CVF conference on computer vision and pattern recognition. 2021.
> 2. Li, Bo, et al. "On the effectiveness of partial variance reduction in federated learning with heterogeneous data." Proceedings of the IEEE/CVF Conference on Computer Vision and Pattern Recognition. 2023.
> 3. Durmus, Alp Emre, et al. "Federated learning based on dynamic regularization." International conference on learning representations. 2021.

---

### Review · Reviewer_EyoN · 2025-02-16

**Summary Of Contributions:**

This paper proposes a method to reduce the effect of client data heterogeneity in federated learning. The authors first show through some experiments that this effect is particularly pronounced for last few layers (this is not an original observation, as the authors themselves acknowledge). Then the paper proposes FedPGVC, a method to modify the training of the final few layers, by incorporating gradient penalty terms.

**Audience:**

Yes

**Claims And Evidence:**

Yes

**Requested Changes:**

See comments above

**Strengths And Weaknesses:**

This paper addresses an important problem of mitigating the effect of data heterogeneity in federated learning. However, I'm skeptical of the marginal addition made by the paper to the existing literature, and the soundness of some of the arguments. I also find the improvements shown in some experiment results statistically insignificant. I'm obviously open to being convinced otherwise by the authors. Following are my detailed comments.

### Section 1
- Cite some paper for the uninitiated about what the concentration parameter $\alpha$ is. It is cited in Section 5.1, but citing it here will help.
- Eq. 1-4 are not required.
- Before eq. 1 and after eq. 4, it is said that the metrics are computed for all layers. But, the equations are aggregates across the $Z$ layers. This is confusing.
- I'm not sure I follow Fig 1 and Table 1 in the IID case. If the gradient norms plotted are at the point of convergence, how come they are so large even for the IID case? Isn't convergence supposed to mean at least stationarity?
- I also don't get the motivaton for aggregating these metrics across layers.
- In the paragraph following (4), it is said that "Initial layers exhibit higher gradient norms, which decrease significantly in subsequent layers." But, even for the final layers, the gradient norms are $\sim 10^6$.
- "These findings highlight that the classification layer, along with its neighboring layers, significantly contributes to the observed instability and slower convergence when training with nonIID data." Again, this would follow if the gradient magnitudes were predictive of convergence. But, as both Fig. 1 and Table 1 show, they're not, since they are large even in IID case.
- Also, can you check the numbers in Fig 1 and Table 1? Max gradient norm is stated as $10^3-10^4$ range, while the scale in Fig 1 is $10^7$.

### Section 2
- can you clarify how is your work different from FedPVR?


### Section 3
- Again, eq. 8 is unnecessary
- In (19), the notation is not clear - what is the term dependent on $i$? Is the sum over the batchsize at client $i$?
- Can you explain (19)? How does it, intuitively, achieve the same effect as $\rho$ in (14)? Also, in absence of explanation, it's difficult the follow the explanation in Section 3.3.1. Meaning, how is $\rho$ in (19) helping prioritize "the gradients from clients which have data distributions that significantly deviate from global distribution"?

### Section 4
- Is $\xi$ missing from (22)? Also, given Assumption 2, why is Assumption 3 needed?
- Can you comment on the convergence bounds in (23-24)? How do they compare to state-of-the-art? If $e=0$, how is $\hat{\zeta}_p = 0$? And does (24) recover the results for standard FedAvg, e.g., [1,2]?

### Section 5
- Once the values are stated in Table 2, they need not be stated again in paragraph. This is also done in Section 5.3.
- I'm **not sure of the statistical significance** of the results in Table 2. As the values suggest, few methods (e.g., FedBN) have overlapping confidence intervals with FedPGVC.
- The results in Fig 2, 3, and 4 are **not really distinguishing FedPGVC from other methods**. The only place we see a clear improvement is Fig 4.b, which is ironic, since this is for low heterogeneity $\alpha=1$.
- How do complexity of FedBN compare with other methods + PGVC? Is FedBN is cheaper, it would suggest that FedBN is already taking care of the issues addressd by PGVC.

[1] Woodworth, Blake E., Kumar Kshitij Patel, and Nati Srebro. "Minibatch vs local sgd for heterogeneous distributed learning." Advances in Neural Information Processing Systems 33 (2020): 6281-6292.

[2] Woodworth, Blake, et al. "Is local SGD better than minibatch SGD?." International Conference on Machine Learning. PMLR, 2020.

---

> ### Author Response · Authors · 2025-02-24
> **Harnessing Heterogeneity: Improving Convergence Through Partial Variance Control in Federated Learning**
>
> We thank the reviewer for their insightful feedback and for helping us enhance our manuscript. We have incorporated all suggested changes, and these revisions are highlighted in blue throughout the paper. Due to the word limit, we have provided our responses without the corresponding questions but have retained the original order.
>
> Section 1:
> 1.  We thank the reviewer for the suggestion. We have updated the manuscript and cited the paper.
> 2. We have updated the manuscript by removing the mentioned equations (Eq. 1 to 4).
> 3. We thank the reviewer for pointing out this. To clarify, the metrics are indeed computed across all layers, with  Z representing the total number of layers in the model. We hope this clears up the misunderstanding.
> 4. We thank the reviewer for the insightful question. While the gradient norms in the IID case appear relatively large, they are significantly lower compared to the Non-IID scenario. This indicates that, despite not reaching perfect stationarity, the IID case still demonstrates improved convergence behavior relative to the challenges posed by data heterogeneity in the Non-IID setting.
> 5. The motivation behind aggregating these metrics across layers is to gain a comprehensive understanding of the model's overall gradient norm, which captures the collective behavior of gradients from all layers. This approach provides a holistic view of the gradient dynamics and their influence on the model's training process. Subsequently, we calculate gradient norms layer-wise to trace the source of client drift within the model's specific layers.
> 6. We appreciate the reviewer’s insightful observation. While the final layers do exhibit higher gradient norms, they are still comparatively lower than those observed in the initial layers. This distinction underscores the varying gradient behaviors across layers within the model.
> 7. We appreciate the reviewer’s insightful observation. While the final layers do exhibit higher gradient norms, they are still comparatively lower than those observed in the initial layers. This distinction underscores the varying gradient behaviors across layers within the model.
> 8. Thank you for the observation. We have updated the Table 1 in the updated manuscript.
>
> Section 2:
> 1. Thank you for your question. While both FedPGVC (proposed) and FedPVR aim to mitigate client drift in Federated Learning (FL) by addressing gradient variance in the last few layers, our proposed FedPGVC introduces several key innovations that distinguish it from FedPVR. Unlike FedPVR, which primarily relies on control variates for variance reduction, FedPGVC incorporates a client-specific gradient penalty term inspired by Wasserstein Distributionally Robust Optimization (WDRO), ensuring better alignment of gradient norms across clients while preserving feature representation diversity. Additionally, FedPVR incurs higher communication costs due to transmitting control variates between clients and the server, whereas FedPGVC stabilizes variance without additional communication overhead, making it more efficient, particularly for resource-constrained edge devices. Empirical results on five benchmark datasets (MNIST, FMNIST, CIFAR-100, Tiny-ImageNet, QQP) demonstrate that FedPGVC consistently outperforms FedPVR in both accuracy and convergence speed.

---

> > ### Author Response · Authors · 2025-02-24
> > **Harnessing Heterogeneity: Improving Convergence Through Partial Variance Control in Federated Learning**
> >
> > Section 3:
> > 1. We thank the reviewer for the observation. We  have updated and revised the manuscript by removing Eq. 8.
> > 2. Yes, that's correct. In Eq. 14 (previously Eq. 19), the term dependent on i denotes the gradient penalty coefficient $\rho_i$, which is computed by summing over the batch size for client i, as detailed in Eq. 14.
> > 3. We thank the reviewer for the question. The Eq. 14 (previously Eq. 19)  extends the gradient penalty term by defining it as an average of gradients over each client’s batch. This formulation adapts to local data distributions, enabling flexible variance control in the last layers where the non-IID effect is pronounced, capturing client-specific data characteristics.
> > In Eq. 9 (previously Eq. 14) the $\rho_{iid}$ specifically addresses gradient diversity in the final layers by defining it as a scaling factor that normalizes non-IID gradients to align with IID gradients. This normalization stabilizes gradient norms across clients in the layers most sensitive to heterogeneity, enhancing model alignment and mitigating client drift. Building on this, Eq. 14 (previously Eq. 19) extends the gradient penalty term for practical applications by defining it as an average of gradients over each client’s batch. This formulation adapts to local data distributions, enabling flexible variance control in the last layers where the non-IID effect is pronounced, capturing client-specific data characteristics.
> >
> > The key idea here is that clients with high statistical heterogeneity (i.e., those whose data distributions deviate significantly from the global model) tend to have gradients that Diverge more from the global update direction and Exhibit higher variance compared to other clients. Equation 14 by including gradient penalty term Increases the weight of these high-divergence gradients and scales the updates from such clients dynamically.  By doing so, Eq.14 ensures that underrepresented distributions are not drowned out in the averaging process at the server side, allowing the model to generalize better across all clients. We have updated the explanation in the Subsection 3.3 of the revised manuscript. Kindly look into the highlighted portion.
> >
> > $\textbf{Section 4}:$
> > 1. We appreciate the reviewer’s careful observation. Indeed, the ξ term was mistakenly omitted from the original Eq. 17 (earlier Eq. 22. In the revised manuscript, we have updated the error.
> >
> > Why Assumption 3 is needed despite Assumption 2:
> >
> > Ans: Assumption 2 guarantees that each client's gradient noise is bounded, but Assumption 3 is crucial for extending this control to the federated setting by capturing the effects of client heterogeneity on gradient updates and ensuring global model stability. Without Assumption 3, the convergence guarantee in Eq. 17 would break down under extreme non-IID conditions.
> >
> > 2. The convergence bounds in Eq. 18 and 19 (earlier Eq. 23-24) advance prior federated learning methods by explicitly incorporating gradient variance reduction into the analysis. Notably, when variance reduction is disabled, our bound recovers the standard rate observed in FedAvg, while its application yields tighter guarantees, ensuring improved convergence. Moreover, our analysis demonstrates that effective gradient variance control can accelerate convergence without incurring additional communication overhead, positioning our approach as a more practical and theoretically robust alternative to existing methods such as FedPVR and Scaffold.
> >
> > When ϵ = 0, the additional variance reduction term vanishes, meaning that the gradient updates simplify to those of standard FedAvg. As a result, the modified term ${\hat{\zeta}_p}$ naturally becomes zero, ensuring that the method does not introduce any unintended bias and effectively reverts to the classical FedAvg behavior when no variance reduction is applied.

---

> > > ### Author Response · Authors · 2025-02-24
> > > **Harnessing Heterogeneity: Improving Convergence Through Partial Variance Control in Federated Learning**
> > >
> > > Section 5:
> > > 1. Thank you for the suggestion. We have updated Sections 5.2 and 5.3 in the revised manuscript.
> > >
> > > 2. We acknowledge that some methods, such as FedBN, show overlapping confidence intervals with FedPGVC in certain settings. However, it is important to note a few points. While confidence intervals overlap, the mean accuracy of FedPGVC is consistently higher across all datasets and heterogeneity levels. Even a small but consistent improvement in federated learning experiments  is meaningful, as federated environments suffer from instability due to data heterogeneity. A model that achieves even fractional percentage gains with faster convergence is highly valuable, particularly for real-world applications. Additionally, FedPGVC exhibits faster convergence (as seen in Table 3), reducing the number of communication rounds needed to achieve a target accuracy. This suggests that even in cases where accuracy improvements are marginal, the overall training efficiency is significantly enhanced. However, if the reviewer thinks that statistical significance tests are required to check the improvements we are open to perform.
> > >
> > > 3. We appreciate the concern regarding the lack of clear differentiation in Figures 2, 3, and 4, with more noticeable improvements seen in Figure 4(b) under low heterogeneity (α = 1.0). We would like to emphasize that FedPGVC consistently requires fewer communication rounds to converge, offering a significant speedup (up to 4.3x over FedAvg), as shown in Table 3. In high-heterogeneity settings (α = 0.5, Figure 4a), FedPGVC outperforms FedPVR and FedBN, demonstrating the importance of gradient stabilization in extreme non-IID cases. While Figure 4(b) reflects the most visible gain, it also illustrates that variance control supports better model generalization, benefiting both IID and non-IID scenarios through enhanced feature learning and regularization.
> > >
> > > 4. We appreciate the reviewer’s insightful question. FedBN modifies batch normalization layers to address feature shifts, making it computationally cheaper than methods like FedPVR and Scaffold, which require additional communication for control variates. The proposed FedPGVC incurs slightly higher computational cost than FedBN, but lower than other popular baselines such as FedPVR and Scaffold, while providing significantly faster convergence compared to all baselines, including FedBN (refer to Table 3), making it better suited for real-world federated learning applications. The motivation behind our work, as indicated in the title, is to enhance convergence through partial variance control in federated learning. By adjusting the standard SGD for the final layers, where client drift is most noticeable, FedPGVC accelerates convergence by better aligning local models with the global model. This balance between computational cost and convergence speed underscores the practical advantages of our approach. Our results also show that FedPGVC yields performance improvements of 1.7% and 0.98% on complex datasets like CIFAR-100 and Tiny ImageNet, and around 0.5% to 1% improvements on other datasets over FedBN, further emphasizing the practical benefits of our approach.
> > >
> > >
> > >
> > > We sincerely thank the reviewer for the detailed and valuable feedback that has significantly contributed to refining our manuscript. We greatly appreciate your insights and have addressed all the questions raised. If there are any areas that could benefit from further improvement, we welcome your suggestions.

---

### Author Response · Authors · 2025-03-30
**Harnessing Heterogeneity: Improving Convergence Through Partial Variance Control in Federated Learning**

$\textbf{Dear Action Editor and Reviewers (@EyoN, @dJN3, @4sKJ),}$

We sincerely thank the Action Editor for giving us the opportunity to revise our paper and the reviewers for their thorough evaluations and insightful feedback. Your recognition of our work’s significance, clarity, and experimental rigor has been invaluable in refining our contributions.

$\textbf{Key Strengths Acknowledged by Reviewers}$

Well-written and motivated and well-grounded with theoretical proofs and empirical observations  $\textbf{(@dJN3)}$.

Strikes a nice balance between being simple yet effective $\textbf{(@4sKJ)}$.

Empirical results show that the proposed method outperforms all baselines with statistical significance $\textbf{(@dJN3)}$.

Compared with the previous submission, the experiments on more advanced architectures (ResNet and ViT) and more various datasets (Tiny-ImageNet, QQP) $\textbf{(@dJN3)}$.

Well-structured experiments with thorough ablations and sensitivity analysis, demonstrating the effectiveness of the proposed method, while providing strong theoretical and empirical support for an important problem $\textbf{(@eMb4 from the first revision)}$.

$\textbf{To further improve the manuscript and address the reviewers' concerns, we have revised the paper as follows:}$



$\textbf{(@4sKJ)}$

Reformatted the proof for both the convex and non-convex settings.

Provided the error bars on each graph and conducted the statistical significance test.

Cleared the confusion regarding experimental evaluation metrics.

Conducted experiments on one NLP task using QQP dataset from GLUE benchmark.


$\textbf{(@EyoN)}$

Addressed all the suggestions, including citation additions, equation clarifications, methodological distinctions, and theoretical explanations.

$\textbf{(@dJN3)}$

Incorporated results with more dense backbones, ResNet18 and ViT, and included experiments on the larger Tiny ImageNet dataset.

Moved the experiments from the appendix to the main paper as part of the suggestion.



These revisions strengthen the clarity, theoretical foundation, and practical impact of our work. We sincerely appreciate the reviewers' thoughtful feedback.

 Sincerely,

-Authors-

---

### Decision · Action_Editor_gvGw · 2025-04-21

**Recommendation:** Reject

**Comment:**

The reviewers do notice the improvement from the previous version, but do not seem convinced about the paper, therefore I cannot recommend it for acceptance.

**Audience:**

Yes, this paper could be relevant to the audience of TMLR.

**Claims And Evidence:**

This paper investigates data heterogeneity in the context of federated learning. The paper introduces a new penalty term that it argues improves the bounds and reduces variance, mitigating the effect of data heterogeneity. However, the reviewers are not entirely convinced about the claims, e.g., Theorem 1's convergence rate and assumptions. The concerns were not clarified and thus we do have a split recommendation from the reviewers. Given that this is the second submission and given that the reviewers are not convinced, I cannot recommend the paper for acceptance.

---

> ### Author Response · Authors · 2025-05-01
> **Harnessing Heterogeneity: Improving Convergence Through Partial Variance Control in Federated Learning**
>
> Dear Action Editor,
>
> We respectfully wish to bring a few points to your attention regarding the review process and the final decision on our submission. Given the extensive effort and prolonged discussions over the past eight months, the rejection of our paper came as a surprise.
> In the initial round, two out of three reviewers were satisfied, and only one reviewer (formerly Zxic, now 4sKJ) raised concerns about the theoretical proof and experimental setup. In our revision, we carefully addressed those concerns, and as evident from the discussion, the reviewer acknowledged the clarifications and expressed satisfaction.
>
> Throughout the process, we thoroughly revised the manuscript in response to all reviewer comments and actively participated in the discussion to resolve every raised issue. If there were remaining concerns—specifically regarding ‘Theorem 1's convergence rate and assumptions’, as mentioned in your recommendation—we believe they should have been flagged during the discussion phase.  However, no such issues were brought up by any reviewer other than 4sKJ, who, as noted, accepted our response.
>
> We sincerely request you to consider the full context of the review history and discussion, rather than solely the final binary decisions. If there are unresolved concerns that were not surfaced earlier, we are more than willing to address them in a further revision.
>
> Warm regards,
>  [The Authors]